# SLE non-coding genetic risk variant determines the epigenetic dysfunction of an immune cell specific enhancer that controls disease-critical microRNA expression

Guojun Hou [1,2,3,4,18], Isaac T. W. Harley [5,6,7,18], Xiaoming Lu [8], Tian Zhou[1], Ning Xu[1], Chao Yao[9], Yuting Qin[1], Ye Ouyang [1], Jianyang Ma[1], Xinyi Zhu[1], Xiang Yu[1], Hong Xu[10,11], Dai Dai[1], Huihua Ding[1], Zhihua Yin[4], Zhizhong Ye[4], Jun Deng[1], Mi Zhou [12], Yuanjia Tang[1], Bahram Namjou [8], Ya Guo[12], Matthew T. Weirauch [8,13,14,15], Leah C. Kottyan [5,8,13,16], John B. Harley[5,8,13,15,17] & Nan Shen [1,2,3,4,8,13✉]

Since most variants that impact polygenic disease phenotypes localize to non-coding genomic regions, understanding the consequences of regulatory element variants will advance understanding of human disease mechanisms. Here, we report that the systemic lupus erythematosus (SLE) risk variant rs2431697 as likely causal for SLE through disruption of a regulatory element, modulating miR-146a expression. Using epigenomic analysis, genome-editing and 3D chromatin structure analysis, we show that rs2431697 tags a cell-type dependent distal enhancer specific for miR-146a that physically interacts with the miR-146a promoter. NF-kB binds the disease protective allele in a sequence-specific manner, increasing expression of this immunoregulatory microRNA. Finally, CRISPR activation-based modulation of this enhancer in the PBMCs of SLE patients attenuates type I interferon pathway activation by increasing miR-146a expression. Our work provides a strategy to define non-coding RNA functional regulatory elements using disease-associated variants and provides mechanistic links between autoimmune disease risk genetic variation and disease etiology.

A list of author affiliations appears at the end of the paper.

The human genome is widely transcribed and most transcripts are noncoding RNAs (ncRNAs)[1,2]. Recent studies have demonstrated that many ncRNAs, such as long ncRNAs (lncRNAs) and microRNAs (miRNAs), play an important role in regulating gene expression in differentiation, development, and human disease[3–8]. Similar to coding genes, ncRNAs transcription is determined by a series of proximal and distal regulatory elements. Understanding the function of ncRNA regulatory elements would shed light on the mechanisms of abnormal ncRNA expression in disease and thus nominate novel therapeutic approaches. However, there are few studies defining the functional regulatory elements of ncRNAs, even for the best-characterized miRNAs.

As the advent of CRISPR/Cas9-mediated sequence perturbation, this method has been considered the gold standard for identifying functional regulatory elements. Numerous studies have adopted CRISPR-based saturation mutagenesis[9], CRISPR interference (CRISPRi)[10–12] or CRISPR activation (CRISPRa)[13] to screen for the functional genomic sequences that regulate coding genes[14–16]. In these screens, libraries containing thousands of single-guide RNAs (sgRNAs) tilling across a broad region of a locus of interest were created and transduced into cells expressing Cas9, dCas9-vp64, or dCas9-KRAB, and protein-level expression or cell proliferation are used as readouts to assess the regulatory effect of noncoding sequences. However, creating CRISPR libraries is laborious, time-consuming, and expensive. For ncRNAs, especially for miRNAs, where the technical challenges in quantifying expression are nontrivial[17], the lack of reliable readouts further impedes performing large-scale screening to define functional noncoding elements.

The human genome harbors thousands of enhancers that shape specific gene expression patterns[18–20]. However, the function and mechanisms underlying cell-specific regulation of most enhancers are largely unknown and little is known about the genes they affect, especially regarding enhancer controlling ncRNAs. Genome-wide association studies (GWASs) have identified numerous disease-associated common variants and the majority of these single-nucleotide polymorphisms (SNPs) reside in noncoding regions of the genome. Thus, they are widely believed to act by altering the function of cell-type-specific DNA regulatory elements[21,22]. This indicates that disease-associated SNPs may act as tags to reveal functional regulatory elements. More importantly, several studies[23–25] have successfully dissected cell-specific enhancers based on this strategy, such as rs356168[25], a Parkinson's disease-associated risk variant located in a distal enhancer of α-synuclein that modulates α-synuclein expression.

Systemic lupus erythematosus (SLE) is a genetically complex autoimmune disease characterized by autoantibody production and dysregulated interferon responses[26–28]. Many ncRNAs are abnormally expressed in this disease[29–31]. Our previous research found that miR-146a expression is significantly downregulated in SLE patients and contributes to the excessive activation of the type I interferon pathway[32]. Meanwhile, treatment with miR-146a mimic can attenuate the pathogenesis of lupus nephritis in the Murphy Roths large (MRL/MpJ)-Fas[lpr]/J (MRL/lpr) lupus mouse model[33]. Although the precise mechanisms leading to decreased miR-146a expression in SLE remain to be determined, genetic variants definitely impact miR-146a expression. For example, rs57095329, a variant in the promoter of miR-146a, influences miR-146a expression by affecting Ets1 binding[34].

To further define the functional disease-associated regulatory elements modulating miR-146a expression and decipher the biological mechanism of miR-146a downregulation in SLE, we analyzed the SLE risk SNPs associated with miR-146a expression, focusing on the rs2431697 locus based on genetic and epigenetic data. rs2431697 lies in an intergenic region between genes encoding pituitary tumor-transforming 1 (PTTG1) and miR-146a. Although several studies nominate miR-146a as the biological effector of this SLE association signal[35], there is limited causal evidence illustrating which gene is responsible for this association. In this study, using genetic, epigenomic, high-throughput screening, and gene-editing approaches in vitro and in vivo, we identify rs2431697 as the likely causal variant in this region by demonstrating that it regulates the miR-146a expression, and that this regulation is cell-type-specific. Furthermore, we demonstrate that the region containing rs2431697 forms a cognate enhancer-promoter loop with the miR-146a promoter and modulates the expression of miR-146a. In particular, the rs2431697 risk variant alters nuclear factor-κB (NF-κB) binding and the chromatin state to fine-tune the expression of miR-146a and its target genes, thus contributing to SLE pathogenesis. Collectively, our strategy exploits information from genetic disease association to identify and define the functional regulatory elements of miR-146a, offering key insights into disease mechanisms.

## Results

**Multi-ancestry fine-mapping and replication studies nominate rs2431697 as a causal risk variant for SLE.** As most disease-associated genetic variants in the noncoding genome have been linked to regulatory elements[21,22], several studies have adopted disease-risk SNPs as a tag to identify key regulatory elements and decipher the mechanism of these SNPs in disease pathogenesis[23–25,36]. However, these studies mainly focus on genetic risk loci encoding proteins rather than ncRNA genes. To test whether disease-associated SNPs could act as regulatory element tags of ncRNAs, we chose miR-146a as our candidate. We did so because miR-146a plays an established, important role in SLE pathogenesis[32] and variants nearby this ncRNA gene have been defined as SLE risk variants by GWAS. We reasoned that the causal variants underlying genetic association with SLE at this locus would likely reveal previously undefined functional regulatory elements for miR-146a.

To begin to define the association signal in the intergenic SLE risk locus between PTTG1-MIR3142HG identified by GWAS[37], we included this locus as part of a larger collaborative fine-mapping effort, the Large Lupus Association Study 2 (LLAS2). We genotyped 74 markers at the PTTG1-MIR3142HG locus in a multi-ancestral discovery cohort, using 12,733 individuals in our final analysis (Supplementary Data 1 and "Quality control and sample overlap" in "Methods" section). We used the Michigan Imputation Server[38] to impute for additional variants using data from the 1000 Genomes project[39]. Altogether, 517 genotyped or imputed variants were used for genetic analysis of this locus with the goal of defining the most likely causal variant to explain the association of this locus with SLE risk.

Logistic regression analysis including admixture estimates as covariates revealed a genome-wide significant association in the cohort of African American (AA) Ancestry (Supplementary Fig. 1a). Further, several intergenic variants associated with SLE risk in this region demonstrated consistent evidence of genetic association ($P < 1E - 4$) across ancestral groups (Supplementary Fig. 1b–d). Trans-ancestral meta-analysis of these association results revealed a robust association of rs2431697 ($P < 1.89E - 22$) (Fig. 1a), with the T allele estimated to increase relative risk by 1.18- to 1.51-fold, depending on the ancestral population (Supplementary Fig. 2). Meta-analysis GWAS studies indicate consistent association of this marker with primary Sjogren's Syndrome, a related autoimmune disease, as well (Supplementary Fig. 3). To define the number of independent genetic effects present at this locus, we performed stepwise conditional logistic regression analysis. As rs2431697 demonstrated the strongest

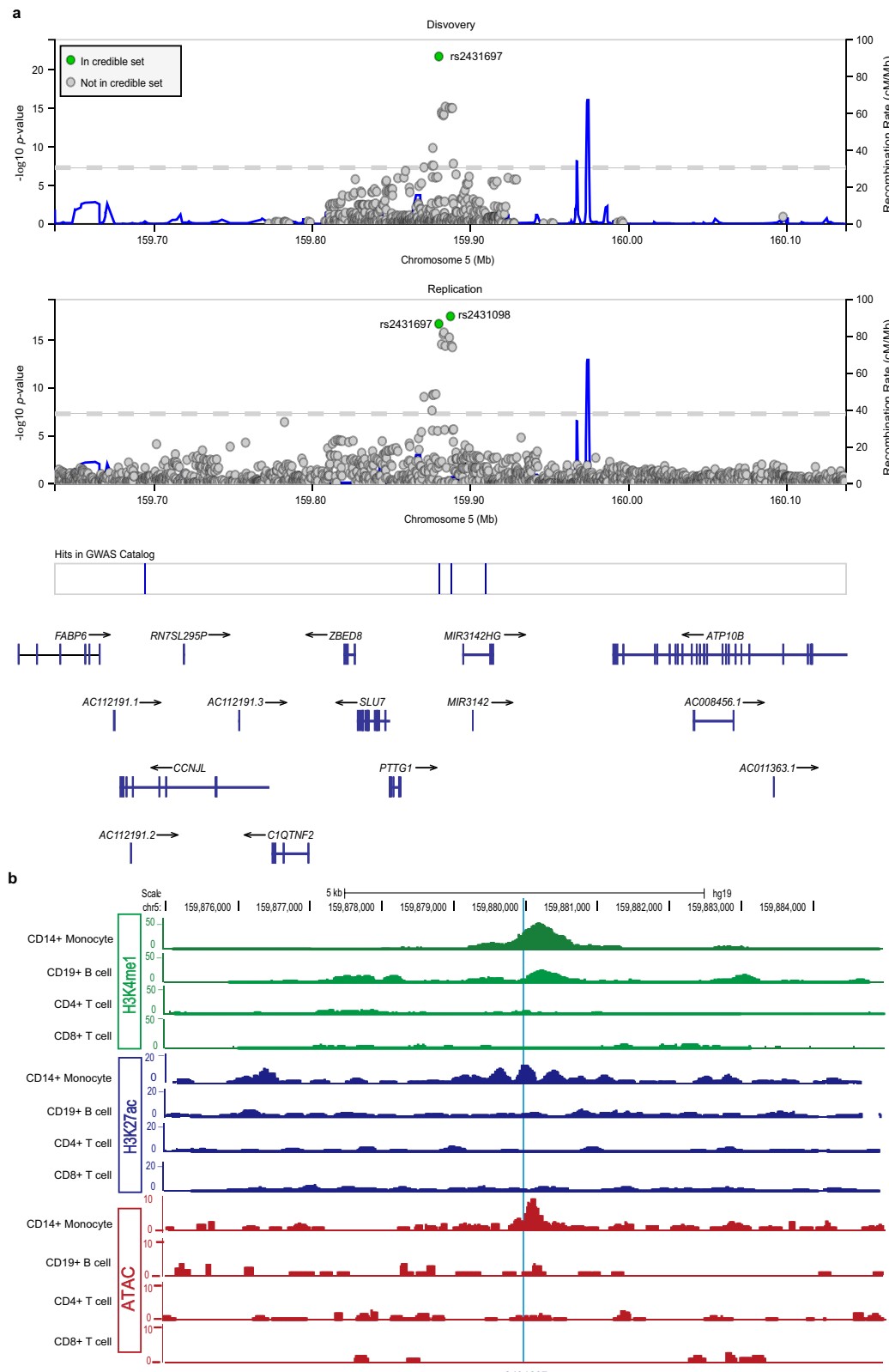

evidence of association with SLE, we first conditioned on this marker in each cohort, under the assumption that this genetic association is due to a shared genetic effect across ancestries. After adjusting for rs2431697 genotype, there were variants with residual association ($1E-2 > P > 1E-4$) present in both the genotyped and imputed data (Supplementary Fig. 4). However, none of the markers were in common across ancestries. Distinct markers, rs2910203 and rs72816340, did account for the residual association signals observed at this locus in the Asian and European ancestry populations, respectively (Supplementary Fig. 4b, c). This analysis supports a model whereby the contributions to SLE risk at this locus independent of rs2431697 are complex, not shared across ancestries and non-substantial relative to that tagged by rs2431697.

**Fig. 1 Genetic and epigenomic analysis of rs2431697. a** *Trans*-ancestral meta-analysis in discovery and replication cohorts identifies 95% credible sets consisting of two markers: rs2431697 and rs2431098. Discovery meta-analysis uses samples in Supplementary Figs. 1 and 5. Replication meta-analysis is from summary statistics of trans-ancestral meta-analysis of summary statistics using the METAL $Z^2$ approach weighted by sample size. Each data point represents a variant in its genomic position (GRCh37/hg19) and the strength of association with SLE. Strength of association is assessed as the −log10 of the P-value. Credible set membership is based on the calculation of Bayes Factors from P-values as implemented in LocusZoom (http://my.locuszoom.org). Variants in green represent members of the 95% credible set, variants in gray represent variants that are not in the 95% credible set. **b** ChIP-seq signal for H3K4me1 and H3K27Ac from Roadmap and ATAC-seq signal detected in our lab reveals a cell-type-specific regulatory element at the rs2431697 locus. chr, chromosome. See also Supplementary Fig. 1–8.

To complement the frequentist approach to genetic analysis of this locus, we similarly performed Bayesian analysis with the genotyped and imputed variants in the *PTTG1-MIR146A* region in each ancestral cohort (Supplementary Fig. 5). We identified a credible set of variants that accounted for 95% of the posterior probability of being causal in this region (1, 1, 23, and 61 genetic variants in the AA, Asian and Asian American (AS), European and European American (EU), and Amerindian (AI) cohorts, respectively). Importantly, a single variant, rs2431697, was common to the 95% credible set across all ancestries. Consistent with the frequentist analysis and the ancestral population-specific linkage disequilibrium (LD) structure at this locus (Supplementary Fig. 6), these results support a shared genetic effect across ancestries mediated by rs2431697.

To confirm the findings from our discovery cohort, we performed *trans*-ancestral meta-analysis based on the summary statistics in a replication cohort of 14,927 individuals[40–42] using the $Z^2$ approach weighted by sample size as implemented in METAL. This replication yielded several genome-wide associated markers and a 95% credible set comprised of rs2431697 ($P = 2.11 \times 1E − 17$, Posterior probability $= 0.136$) and rs2431098 ($P = 3.39 \times 1E − 18$, Posterior probability $= 0.826$) (Fig. 1a). The inclusion of rs2431697 in the 95% credible set of replication cohort is consistent with our finding that this marker (or a marker in LD with it, such as rs2431098) likely represents the causal variant at this locus.

**eQTL analysis identifies rs2431697 as a regulator of miR-146a expression**. Because of its proximity to miR-146a, a gene previously implicated in SLE biology[32], we reasoned that rs2431697 might constitute an expression quantitative trait locus (eQTL) for miR-146a. Indeed, a previous study by Pickrell et al.[43] defined rs2431697 as the variant most strongly correlated with miR-146a expression across the entire genome. This finding is in agreement with a study finding that rs2431697 genotype correlates with miR-146a expression, but not the adjacent neighboring gene *PTTG1* in SLE patients and controls[35]. The Pickrell et al.[43] study examined gene expression levels from RNA sequencing (RNA-Seq) analysis of lymphoblastoid cell lines (LCLs) derived from Yoruban ancestry individuals from Ibadan, Nigeria (hereafter referred to as YRI) generated as part of the International Haplotype Map (HapMap) project[44]. As many of the samples overlapped with the 1000 Genomes Project[39], we re-analyzed the gene expression data using the comprehensive genotype data from the 1000 Genomes project[45]. Consistent with the report by Pickrell et al.[43], several intergenic variants in this region were associated with miR-146a expression. Among these, rs2431697 was most strongly associated ($P = 4.95E − 5$) (Supplementary Fig. 7a). An increasing number of the SLE-protective rs2431697-G allele was strongly correlated with increased miR-146a expression, consonant with a prior finding of increased miR-146a expression in peripheral blood mononuclear cells (PBMCs) of normal control subjects in comparison to SLE patients and our genetic data (Supplementary Fig. 7a)[32]. Conditional logistic regression analysis including the genotype of rs2431697 as a covariate abrogated

the association signal in this intergenic region (Supplementary Fig. 7b). Furthermore, although expression of the other genes in this region was nominally associated with the genotype of variants adjacent to the respective promoters of several genes in the region, there was no substantive overlap between the intergenic SLE risk association variants and the expression of any other gene in the region (Supplementary Fig. 8). Importantly, rs2431697 genotype did not associate with the expression of any other gene across the entire genome in this study at the $P < 1E − 4$ threshold. Taken together, our genetic association and eQTL analyses strongly support rs2431697 as the causal variant mediating SLE risk at the *PTTG1-MIR3142HG* intergenic locus with effects on miR-146a gene expression as the biological explanation for this association signal.

**MPRA screening and publicly available data nominate the rs2431697-containing region as a potential cell-type-specific enhancer**. Our massively parallel reporter assay (MPRA), which was used to screen allelic enhancer activity for all genome-wide significant SLE-associated genetic variants, discovered rs2431697 is a functional SNP with significant genotype-dependent enhancer activity (Supplementary Data 2)[46]. Likewise, using publicly available epigenetic data (NIH Roadmap Epigenomics Consortium, http://www.roadmapepigenomics.org)[47], we observed strong H3K4me1 and H3K27ac signals, histone modifications that are hallmarks of active enhancers, overlapping rs2431697 in CD14+ monocytes. In CD19+ B cells, we only found H3K4me1 modifications in this region, indicating that this region is likely a poised enhancer in B cells. However, there were no detectable H3K4me1 and H3K27ac signals at this locus both in CD4+ T cells and CD8+ T cells (Fig. 1b). Another feature of active enhancers is high chromatin accessibility. To test the chromatin accessibility at rs2431697 locus, we performed assay for transposase-accessible chromatin using sequencing (ATAC-seq) in human primary immune cell subsets. Consistent with the enhancer-associated histone marks, our ATAC-seq assays showed that this chromatin region is only open in CD14+ monocytes (Fig. 1b). Taken together, these data indicate that the rs2431697-containing region may be a cell-type-specific enhancer regulating nearby gene expression. Based on the above data, we focused the remainder of our study on rs2431697.

**A functional enhancer at rs2431697 specifically modulates miR-146a expression**. Functional enhancers can be identified by CRISPRi or CRISPRa[10,13,48]. To test whether this region is a functional enhancer regulating miR-146a expression, we carried out CRISPRi and CRISPRa experiments with sgRNAs targeting the rs2431697 locus in the U-937 cell line. Consistent with our hypothesis, CRISPRa and CRISPRi led to robust increased and decreased expression of miR-146a, respectively (Fig. 2a, c). Importantly, the expression of nearby genes within a 2 Mb region of rs2431697 was not affected in the assays (Fig. 2b, d).

To directly assess the role of the rs2431697-containing region in regulating miR-146a expression, we deleted a 30 bp region at

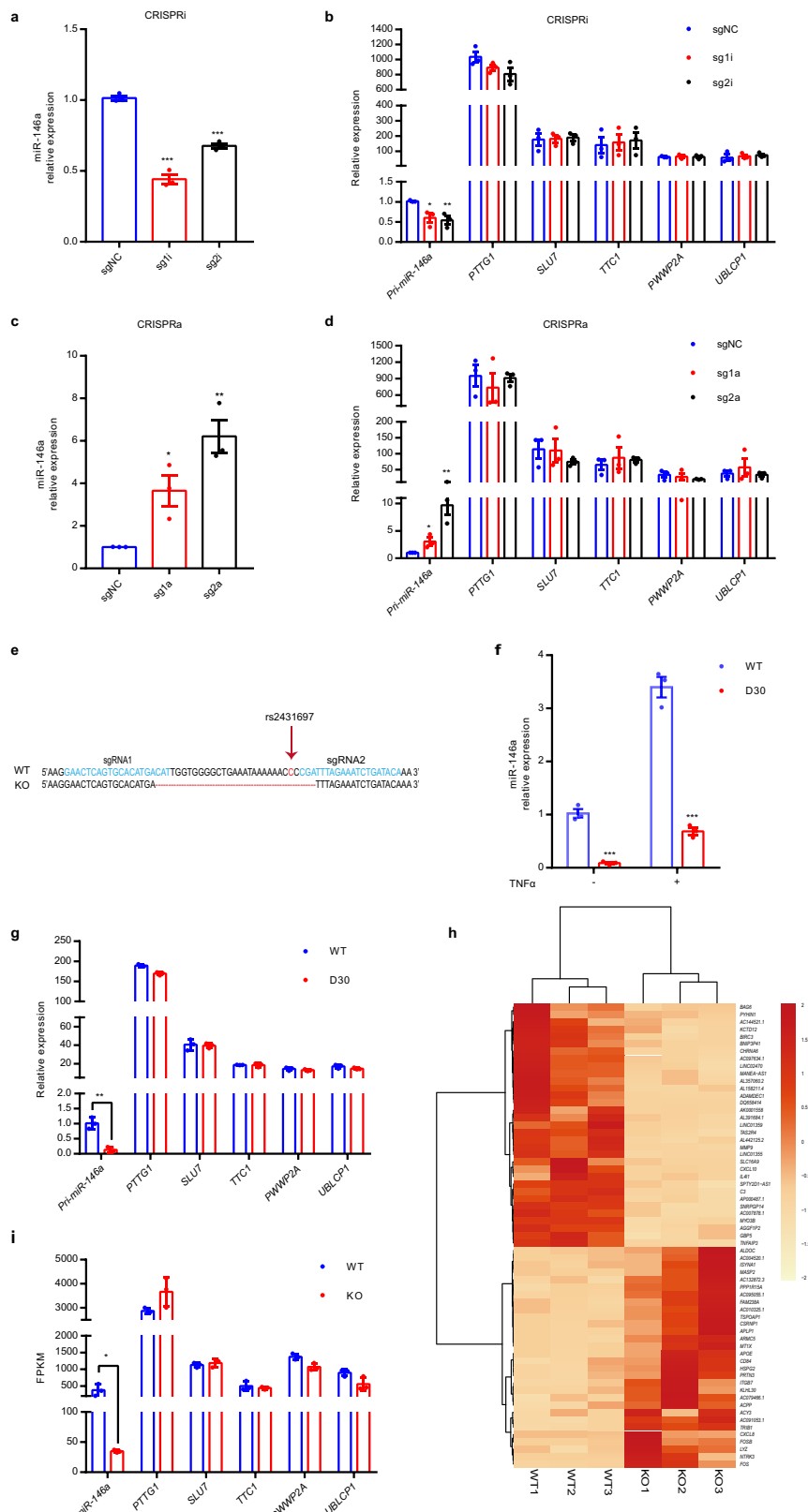

rs2431697 in the U-937 cell line using CRISPR/Cas9 with two sgRNAs targeted around the rs2431697 site (Fig. 2e). Plasmids expressing Cas9 and sgRNAs were nucleofected into cells and cell clones were screened to obtain bi-allelic 30 bp deletion (D30) clones. Clones exposed to CRISPR/Cas9 complexes with wild-type (WT) genotype were selected as controls. As shown in

Fig. 2f, the deletion of the 30 bp region harboring rs2431697 resulted in lower expression of miR-146a compared to the WT clones in both unstimulated and tumor necrosis factor-α (TNFα)-stimulated conditions. We also examined the expression of other genes within the 2 Mb region surrounding rs2431697 and found no significant difference between the WT clone and D30 clone

**Fig. 2 CRISPRi, CRISPRa, and CRISPR/Cas9-mediated genome deletion demonstrate that the rs2431697-containing region is a functional enhancer that specifically regulates miR-146a expression. a–d** RT-qPCR analysis of the expression of miR-146a and nearby genes within the 2 Mb region of rs2431697 in U-937 cells. sgRNAs targeting the rs2431697-containing region were used in CRISPR interference (CRISPRi) (**a**, **b**) (miR-146a sg1i: ***$P <$ 0.0001, miR-146a sg2i: ***$P = 0.001$, *Pri-miR-146a* sg1i: *$P = 0.0215$, *Pri-miR-146a* sg2i: **$P = 0.0096$) and CRISPR activation (CRISPRa) (**c**, **d**) (miR-146a sg1a: *$P = 0.0227$, miR-146a sg2a: **$P = 0.0027$, *Pri-miR-146a* sg1a: *$P = 0.0428$, *Pri-miR-146a* sg2a: **$P = 0.0065$) assays ($n = 3$, biological replicates). **e** Deletion of a 30 bp fragment containing rs2431697 with the CRISPR/Cas9 technology. **f**, **g** RT-qPCR analysis of the expression of miR-146a and nearby genes within 2 Mb of rs2431697 in U-937 wild-type clones and deletion clones (miR-146a without TNFα stimulation: ***$P = 0.0004$, miR-146a with TNFα stimulation: ***$P = 0.0002$, *Pri-miR-146a*: **$P = 0.0018$) ($n = 3$, biological samples replicates). **h** Heat map of differentially regulated genes within the rs2431697-containing region as measured by RNA-seq (log2 fold-change > 1.5 and FDR < 0.05). The red and orange colors in the heat map depict higher and lower gene expression, respectively. The color intensity indicates the magnitude of the expression differences. **i** RNA-seq analysis of the expression of genes within 2 Mb of rs2431697 in U-937 wild-type clones and deletion clones, *$P = 0.0306$ ($n = 3$, biological samples replicates). WT: rs2431697 wild type, D30 or KO: 30 bp fragment-harboring rs2431697 deletion. Data are represented as mean ± SEM and $P$-values are calculated using unpaired two-tailed Student's $t$-test. *$P < 0.05$; **$P < 0.01$; ***$P < 0.001$.

(Fig. 2g). In addition, we performed RNA-seq on three WT clones and three deletion clones to identify *cis-* and *trans-*regulatory effects of region containing rs2431697. Among the analyzed 13,372 transcripts, we observed 63 genes with significant differential expression with a false discovery rate (FDR) cutoff ≤ 0.05 and a log2 fold-change of ≥1.5 (Fig. 2h and Supplementary Data 3), and miR-146a was the only influenced genes within the 2 Mb region of rs2431697 (Fig. 2i). Next, we analyzed the enriched pathways of these differentially expressed genes in KEGG (Kyoto Encyclopedia of Genes and Genomes) and found the top pathway to be the TNF signaling pathway. These genes also enriched in the interleukin-17 (IL-17) and NF-κB signaling pathways (Supplementary Data 3), both of which are consistent with the function of miR-146a. Altogether, these data indicate that the genomic region harboring rs2431697 is a functional enhancer specifically regulating miR-146a expression.

**In vitro and in vivo studies reveal the regulatory element at rs2431697 as a cell-type-specific enhancer regulating miR-146a expression.** Enhancers are often cell-type-specific and regulate target gene expression in a cell-context-dependent manner. The above data demonstrate that a regulatory element at rs2431697 can functionally regulate miR-146a expression in U-937 cells. To define the cell-type specificity by which the rs2431697-containing region regulates miR-146a expression, we also generated a 30 bp deletion clone in Raji and Jurkat cells. In Raji cells, we observed similar results to those in U-937 cells (Fig. 3a–c). However, loss of this region has little effect on miR-146a expression in Jurkat cells (Fig. 3b–d). In U-937 and Raji cells, this SNP-containing region is an active enhancer marking with high enrichment of H3K4me1 signal, H3K27ac signal, and high chromatin accessibility (Fig. 3e, f). In Jurkat cells, there were scant signals for enhancer marks (Fig. 3e, f). These data suggest that this regulatory element functions in concert with the observed epigenetic modifications.

To better define the cell type-specific regulation in the rs2431697-containing region, we isolated three major immune cell types from the PBMCs of five healthy donors including CD14+ monocytes, CD3+ T cells, and CD19+ B cells. Following isolation, the rs2431697-containing region was edited by electroporation of Cas9 RNP with flanking sgRNAs in these primary immune cells (Fig. 3g). The editing efficiency of the target locus was first estimated by T7EI assay (Supplementary Fig. 9a–c) and samples with more than 40% editing efficiency were used for analysis. In agreement with the epigenetic modifications observed in the different immune cell types, disruption of rs2431697-containing region significantly decreased miR-146a expression in CD14+ monocytes (Fig. 3h), but not in CD3+ T cells or CD19+ B cells (Fig. 3i, j).

Next, we sought to validate these findings in vivo. As numerous enhancers are not well-conserved or are fundamentally different between human and mice, we reasoned that employing murine model systems to define the function of this regulatory element would pose substantial problems with interpretation. Therefore, we focused our efforts on defining the function of this human gene enhancer within human cells in vivo. We first constructed a humanized mouse model by delivering human whole PBMCs to NOD-*scid* IL2Rγ^null (NSG) mice. After transplant for 24 days, Adenovirus expressing Cas9-GFP and rs2431697 target dual sgRNAs or negative control sgRNAs were injected intraperitoneally. 3 days after injection, 100 human CD14+ monocytes, CD3+ T cells, and CD19+ B cells expressing green fluorescent protein (GFP) were isolated by fluorescence-activated cell sorting (FACS), respectively (Fig. 3k and Supplementary Fig. 9d). Using miScript Single Cell qPCR Kit, miR-146a expression was quantified in isolated cells. Consistent with the in vitro data, disruption of the genomic region harboring rs2431697 only reduced miR-146a expression in monocytes (Fig. 3l–n). Altogether, our in vitro and in vivo data imply that the rs2431697-containing region harbors a functional cell-specific enhancer regulating miR-146a expression in primary human cells in vivo.

**A functional enhancer-promoter connection exists between the regulatory element at rs2431697 and the miR-146a promoter.** Distal enhancers usually affect gene expression by forming cognate enhancer-promoter loops with the target gene promoter site. Previous studies have defined two promoters for *MIR3142HG*, the gene encoding miR-146a, based on luciferase reporter assays[34,49,50]. There is a distal promoter located upstream of the first exon of *MIR3142HG* that contains an NF-κB-binding site (Fig. 4a, b). There is also a proximal promoter located upstream of the *MIR3142HG* second exon, which contains a PLZF-binding site (Fig. 4a, c). We utilized the CRISPR/Cas9 technology to separately delete the NF-κB-binding site sequence and the PLZF-binding site sequence (Fig. 4b, c). We found that NF-κB-binding site deletion led to dramatically decreased miR-146a expression in both the unstimulated and TNFα-stimulated conditions (Fig. 4d). However, the PLZF-binding site deletion did not significantly influence miR-146a expression (Fig. 4e). These data reveal that the distal promoter initiates *MIR3142HG* transcription in U-937 cells, leading to subsequent processing and expression of miR-146a.

After verifying the promoter region of miR-146a, we carried out circularized chromosome conformation capture sequencing (4C-seq) assays to measure the three-dimensional chromatin topology of this region in these cells. As shown in Fig. 4f–h, regardless of whether the view point is oriented to the rs2431697 region or the miR-146a promoter region, the interaction between the two regions exists. These data establish a physical connection between the rs2431697 region and the miR-146a promoter that could provide a

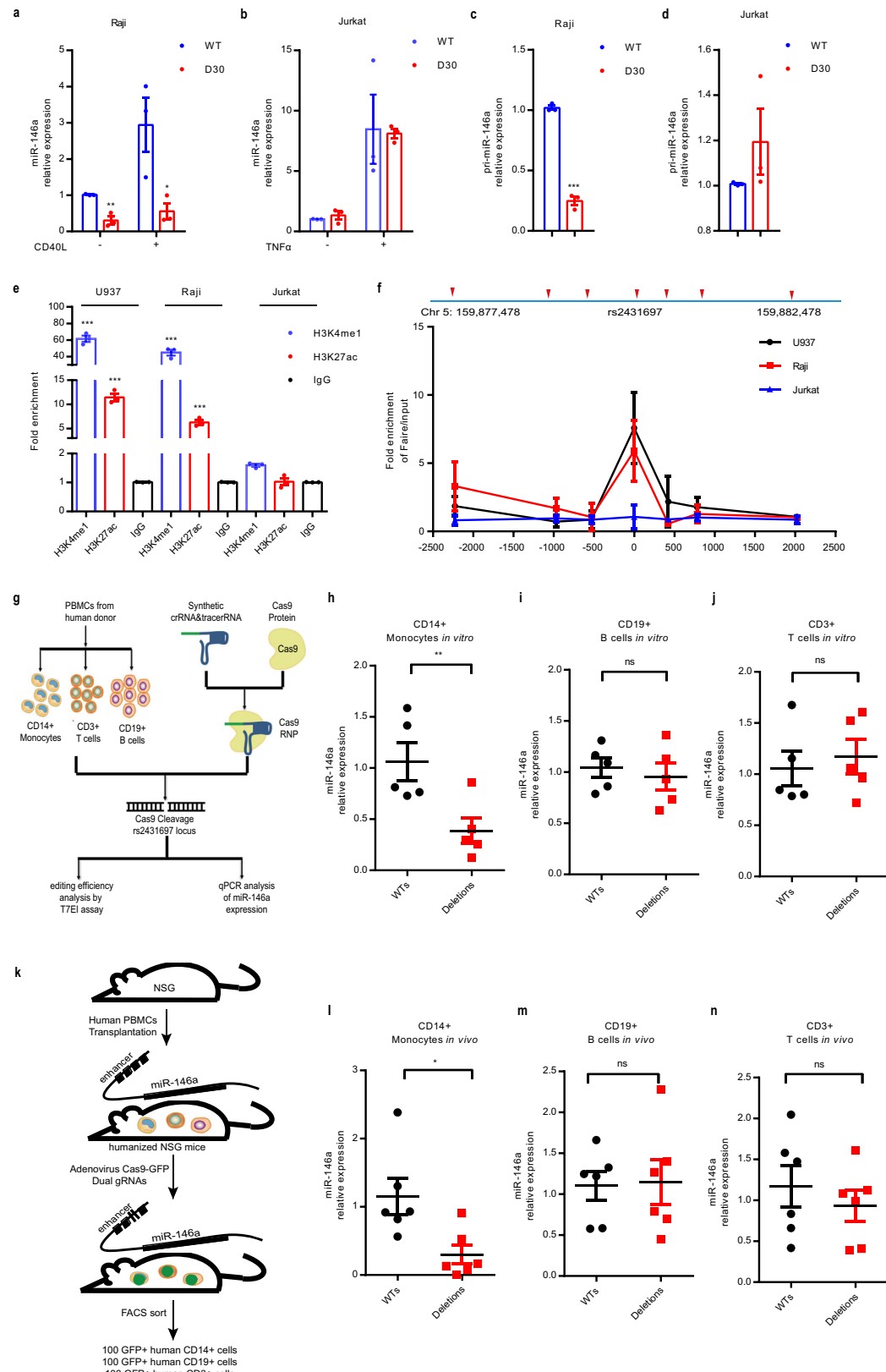

mechanistic explanation for the modulation of miR-146a expression by the regulatory element in the rs2431697 region.

**rs2431697 alleles differentially regulate miR-146a expression by modulating transcription factor binding and chromatin state**. To test whether the effect on miR-146a expression is

mediated by the genotype of rs2431697, we generated cell clones harboring either the homozygous major (T/T) allele or the homozygous minor (C/C) allele using CRISPR/Cas9-induced homologous recombination (Fig. 5a). After screening more than 200 clones, four clones of each genotype with evidence of successful homologous recombination were obtained. For each

**Fig. 3 The genomic region harboring rs2431697 is a cell type-specific enhancer regulating miR-146a expression. a–d** RT-qPCR analysis of miR-146a expression in Raji or Jurkat WT and deletion clones (**P = 0.0029, *P = 0.0381, ***P < 0.0001) (n = 3, biological replicates). WT: rs2431697 wild type, D30: 30 bp fragment-harboring rs2431697 deletion. **e** ChIP-qPCR analysis of active enhancer marks (H3K4me1 and H3K27ac) within the rs2431697-containing region in U-937, Raji, and Jurkat cells (U-937 H3K4me1: ***P < 0.0001, U-937 H3K27ac: ***P = 0.0002, Raji H3K4me1: ***P = 0.0003, Raji H3K27ac: ***P = 0.0004) (n = 3, biological replicates). Fold enrichment was calculated as the ratio of the signal from the target antibody sample to the signal from the IgG control as measured by qPCR. **f** FAIRE-qPCR analysis of open chromatin and nucleosome occupancy for a rs2431697-centered 5 kb region of the 5q33.3 locus in U-937, Raji, and Jurkat cells (n = 3, biological replicates). Fold enrichment was calculated as the ratio of the signal from the FAIRE sample to the signal from the input control DNA as measured by qPCR. Primer sets and amplicons are indicated by a red arrow. Chr, chromosome. **g** Experimental scheme of Cas9 RNP delivery to different primary immune cells for genome editing and phenotypic characterization. **h–j** RT-qPCR analysis of miR-146a expression in CRISPR/Cas9 RNP edited CD14+ monocytes (**h**) (**P = 0.0059), CD19+ B cells (**i**) and CD3+ T cells (**j**) isolated from donor PBMCs (n = 5, biological samples replicates). **k** Schematic illustration of the construction of humanized mice to study cell type-specific regulation. NSG, NOD-scid IL2Rγ^null. **l–n** In vivo editing of the rs2431697-containing region in humanized mice and RT-qPCR analysis of miR-146a expression in edited CD14+ monocytes (**l**) (*P = 0.0195), CD19+ B cells (**m**), and CD3+ T cells (**n**) (n = 6, biological mouse replicates). Data are represented as mean ± SEM and P-values are calculated using unpaired two-tailed Student's t-test (**a–e**, **l–n**) and paired two-tailed Student's test (**h–j**). *P < 0.05; **P < 0.01; ***P < 0.001. See also Supplementary Fig. 9.

genotype (T/T and C/C), four clones were used to detect the effect of rs2431697 genotypes on miR-146a expression under both resting and TNFα stimulation conditions. As shown in Fig. 5b, resting C allele clones exhibited increased expression of miR-146a relative to T-allele clones, which is exacerbated in the setting of TNFα stimulation, a known stimulus of NF-kB signaling in these cells. These data are consistent with the described role of miR-146a as a negative regulator of immune responses, the protective role of the C allele in SLE, and the eQTL data above correlating increased miR-146a expression with genetic dose of C alleles.

We next sought to explore the potential mechanism of rs2431697 allele-specific regulation. Given that SNPs in regulatory regions usually function by influencing transcription factor (TF) binding, we performed DNA-affinity precipitation assay (DAPA) followed by mass spectrometry (MS), to identify the possible candidates that specifically bind to the 41 bp nucleotide sequence containing rs2431697 using U-937 cell nuclear extracts. Compared to random DNA sequence, the unique TFs enriched by the nucleotide sequence harboring rs2431697 are mainly NF-κB family proteins (Supplementary Fig. 10a). We also observed PU.1 binding (Supplementary Fig. 10a and Supplementary Data 4). Bioinformatic analysis using HaploRegV4.1, JASPAR, and Human TFs catalog[51] revealed overlap between rs2431697 and a slightly truncated binding motif for NF-κB (Fig. 5c), whereas the PU.1-binding sites are located adjacent and do not directly overlap with rs2431697 (Supplementary Fig. 10b, c). Remarkably, the rs2431697 T risk allele exhibits lower binding affinity for the predicted NF-κB-binding motif relative to that of the rs2431697 C non-risk allele (Fig. 5c). To evaluate the effect of this variant on regulator binding, we carried out electrophoretic mobility shift assays (EMSAs) using nuclear extracts from HEK-293T cells over-expressing RELA, an NF-kB subunit, along with probes for the risk allele and the non-risk allele of rs2431697. Strong genotype-dependent binding of RELA was observed, with a preference for the protective C allele at rs2431697 (Fig. 5d). This observation was further validated through RELA chromatin immunoprecipitation followed by quantitative PCR (ChIP-qPCR) using a CRISPR/Cas9-mutated U-937 clone that was heterozygous for rs2431697, specifically the anti-RELA ChIP identified enhanced binding of RELA to the chromosome containing the C allele compared to the chromosome containing T allele (Fig. 5e).

As a causal SNP could alter chromatin accessibility via the different binding of a TF, we adopted formaldehyde-assisted isolation of regulatory element (FAIRE) allele-specific qPCR (AS-qPCR) to examine the risk allele and non-risk allele FAIRE signal in this rs2431697 heterozygous cell clone. The results indicate that the fragment harboring the rs2431697 C non-risk allele

exhibited more FAIRE signal than the fragment harboring the rs2431697 T risk allele (Fig. 5f). In addition, ATAC-Seq analysis of monocytes revealed that the reads of fragments containing the C allele are much more abundant than fragments containing the T allele (Supplementary Fig. 10d) in heterozygous samples. We also examined the allelic distribution of H3K4me1 and H3K27ac at the rs2431697 site using public ChIP-seq data to assess allelic bias using the MARIO pipeline[52] and found that this SNP showed significant bias in the non-risk allele direction for these marks as well (Supplementary Fig. 10e). From the above data, we conclude that the DNA sequence containing the rs2431697 C non-risk allele binds NF-κB with higher affinity and has greater accessibility relative to the rs2431697 T risk allele, thus driving increased expression of miR-146a.

**Targeting the enhancer region of rs2431697 attenuates the SLE IFN score by upregulation of miR-146a expression.** As a key negative regulator of the type I interferon pathway, miR-146a is an attractive therapeutic target in SLE[53], in part due to its activity in targeting SLE risk genes within or upstream of the type I interferon signaling pathway, such as IRF5, TRAF6, IRAK1, and STAT1[32,50]. Taken together, evidence implicating the rs2431697-containing region as a critical regulator of miR-146a expression and our prior observation that miR-146a expression is decreased in the PBMC of SLE patients suggested a novel therapeutic approach for SLE. We hypothesized that restoring miR-146a expression to SLE PBMC would restrain the high type I interferon score/signature seen in some SLE patients (High IFN Score). Furthermore, given the allele-specific enhancer function of the rs2431697-containing region, we reasoned that miR-146a modulation and secondary effects might be achieved by directly targeting this enhancer. To assess the feasibility of this novel therapeutic approach, we used the CRISPRa system to activate miR-146a expression with sgRNAs targeting the rs2431697 locus. If our hypothesis is correct, then this intervention should inhibit type I interferon pathway activity, if employed in the PBMC of High IFN Score SLE patients. Importantly, this would constitute a plausible direct mechanistic link between this GWAS association signal and SLE etiopathogenesis, given the important role that type I interferon (IFN) production and signaling plays in this disease[53]. We first chose six WT clones and six rs2431697 deletion clones to test the effect of the rs2431697-containing region on interferon-inducible gene expression. These clones were first treated with Interferon-α, and interferon-inducible gene expression was quantified. qPCR data indicate that deletion clones were more sensitive to interferon-α stimulation than WT clones, as reflected by increased expression of interferon signaling

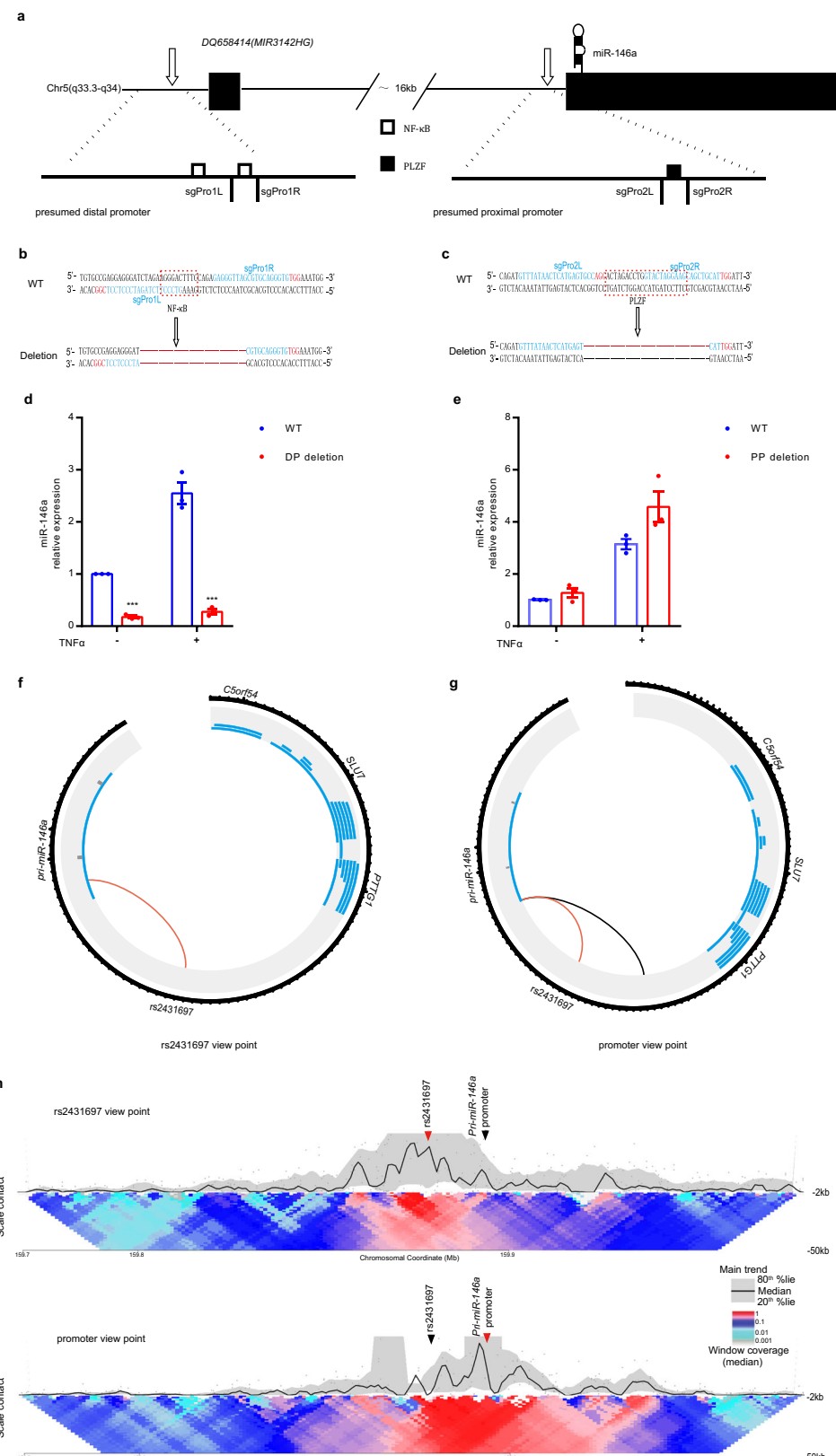

responsive genes *IFIT1*, *IFIT3*, *OAS1*, and *IFI44* (Fig. 6a–d), suggesting that the SLE risk mediated by the rs2431697-containing region may well act through miR-146a-target gene regulation within the type I interferon pathway. Next, we isolated PBMCs from SLE patients with high IFN scores and performed a CRISPRa experiment in these PBMCs (Fig. 6e). As expected,

CRISPRa upregulated *pri-miR-146a* expression (Fig. 6f) and downregulated the IFN score (Fig. 6g), which is an important standard to assess IFN activity in SLE. Taken together, these data indicate that targeting this enhancer may be an effective method to modulate the IFN pathway, which is highly relevant to SLE etiopathogenesis.

**Fig. 4 The rs2431697-containing region forms looping interactions with the miR-146a promoter. a** The position of the distal promoter and proximal promoter in the genome. Chr, chromosome. **b, c** Genotype, sgRNA position, and transcription factor-binding sequence of wild-type and mutated promoters. **d, e** RT-qPCR analysis of miR-146a expression in U-937 promoter WT clones and promoter deletion clones (without TNFα stimulation: ***$P < 0.0001$, TNFα stimulation: ***$P = 0.0004$). DP, distal promoter, PP, proximal promoter, ($n = 3$, biological replicates). **f, g** Circos plot visualizing significant *cis* interactions. The black circle represents the 100 kb region around rs2431697 (**f**) and the 130 kb region around the miR-146a promoter (**g**). For each gene in this region, transcripts are depicted (without the exon/intron details) in blue. For the rs2431697 view point, connections between the rs2431697 locus and interacting regions within 5 kb of any TSS are displayed. For the miR-146a promoter view point, connections between the miR-146a promoter site and rs2431697 nearby interacting regions are displayed. Red line indicated the interactions between the rs2431697 locus and miR-146a promoter. **h** Contact profiles of the rs2431697 SNP site (top panel) and *Pri-miR-146* promoter (bottom panel) using a 2 kb window size in main trend subpanel (black line). Red arrow head indicated view point position, black arrow head indicated the target position. Gray dots indicate normalized contact intensities and gray band shows the 20–80% percentiles. Heat map displays a set of medians of normalized contact intensities calculated at different window sizes (from 2 to 50 kb), the red and blue colors in the heat map depict higher and lower interaction intensity, respectively. Data are represented as mean ± SEM, and *P*-values are calculated using unpaired two-tailed Student's *t*-test. *$P < 0.05$; **$P < 0.01$; ***$P < 0.001$.

## Discussion

In this study, we describe a strategy to identify the functional regulatory elements controlling the expression of a ncRNA by integrating genetic data, epigenetic data, and gleaning mechanistic insight using genome editing with CRISPR. We decipher the biological mechanism that mediates the risk for SLE conferred by rs2431697 and demonstrate that it likely alters SLE pathogenesis by regulating miR-146a expression (Fig. 7).

The pattern of miRNA expression is tissue and disease specific[54–56]. The specificity of this expression pattern is shaped by super-enhancers, which consist of multiple regulatory elements[57]. Defining the correct disease-associated regulatory elements will increase our understanding of the mechanisms that direct distinct miRNA expression profiles in disease-relevant cell subpopulations. Disease-risk SNPs in noncoding sequences can mark functional regulatory elements. Our genetic study and high-throughput screening identify rs2431697 as the putative causal SLE risk variant explaining the observed association signal. Furthermore, this variant's correlation with miR-146a expression as well as epigenetic analysis suggests that this genetic variant is located in a distal and cell-specific enhancer of miR-146a. CRISPR-mediated fragment deletion, CRISPRi, and CRISPRa studies in vitro and in vivo define this region as a functional enhancer, specifically regulating miR-146a expression in a cell type-dependent manner. This suggests that many other intergenic disease-associated genetic variants may also act as regulators of functional enhancers of ncRNAs, such as miRNAs, which are capable of fine-tuning cellular responses by altering the expression of many genes in *trans*.

CRISPR/Cas9-mediated fragment deletion is the gold standard to define the identity of genomic regions as functional enhancers. In our study, we deleted the rs2431697-containing region in different cell lines and corresponding primary immune cells to define the cell-type specificity of this enhancer. We observed that the deletion of this region downregulated miR-146a expression in U-937 and Raji cell lines but not in the Jurkat cell line. However, in primary immune cells, disruption of this enhancer only influences miR-146a expression in CD14+ monocytes. This result is consistent with the epigenetic modification around rs2431697 and the disparities suggest that cell lines cannot always fully mimic the context of primary cells. The most informative answer to the question of whether a genetic variant alters the function of a cell type-specific enhancer in primary human cells necessarily involves the use of primary human cells. Our data clearly define an enhancer with specific activity in myeloid cells. However, given the findings that the rs2431697-containing region is a poised enhancer in B cells, and that its effect on modulating expression is enhanced in myeloid cells upon stimulations, future studies of the effects of this region in primary B cells during other disease-relevant stimulus conditions, such as downstream of B-

cell receptor signaling or in the context of Epstein–Barr virus infection are warranted.

Confirming that the rs2431697-containing region does not operate in disease-relevant contexts in primary CD19+ B cells will be important, in light of the key role of autoreactive B-cell clones in SLE and the discrepant results in primary CD19+ B cells relative to those observed in a B-cell line (Fig. 3 a, d, e, I, m).

Distal enhancers act by spatially approximating the cognate promoter region to regulate target gene expression[58–60]. Our 4C-seq experiment confirmed a physical interaction between the rs2431697 locus and the miR-146a promoter region, which provides complementary evidence regarding the functional role of rs2431697-containing region. We also observed that other promoter regions connect with the rs2431697-containing region, such as the *TIMD4* promoter, the *PROP1* promoter, etc. (Supplementary Data 5). However, these genes are expressed at very low levels in monocytes, indicating that transcription of these genes is likely controlled with additional looping interactions. Importantly, studies have also shown that chromatin loops can occur at non-activated genes[61]. miR-146a expression is controlled by multiple proximal and distal regulatory elements. We also found other genomic regions that connect with the miR-146a promoter, but whether these regions also represent functional enhancers remains unclear. Dissecting these potential regulatory elements will help to provide a more complete picture of miR-146a transcriptional regulation.

eQTL analysis indicates that rs2431697 is associated with miR-146a expression. We confirmed this using CRISPR/Cas9-mediated homology repair to generate different rs2431697 allele-harboring clones and validated the allele-specific regulation of miR-146a expression. Using bioinformatic analysis and experimental approaches, we discovered that NF-κB is the key protein differentially binding to the rs2431697 risk allele and non-risk allele. Through this differential binding, NF-κB likely modulates the differential regulatory function of rs2431697 between the risk and non-risk alleles. Interestingly, we also found PU.1 enriched in the DNA sequence nearby rs2431697. PU.1 and CEBPA are considered important lineage-determining TFs that acts to prime cell-specific enhancers[62] and bioinformatic analysis indicates that the binding sequences of PU.1 and CEBPA are found surrounding rs2431697 (Supplementary Fig. 10c–d). This observation, to some extent, might help explain why the rs2431697 locus is a cell-type-specific enhancer. Although DAPA-MS study reveals the binding of NF-κB or PU.1 and DNA oligo harboring rs2431697, a 41 bp DNA sequence can pull down more than 100 proteins in our study, these proteins include histone proteins, ribosomal proteins, chromatin structure maintenance-related proteins, TFs and so on (Supplementary Data 4). It is really a challenge to distinguish the real binding and functional proteins. There is urgent need to develop a novel and precise technology to identify the genetic varitant binding proteins. In addition to differential TF-

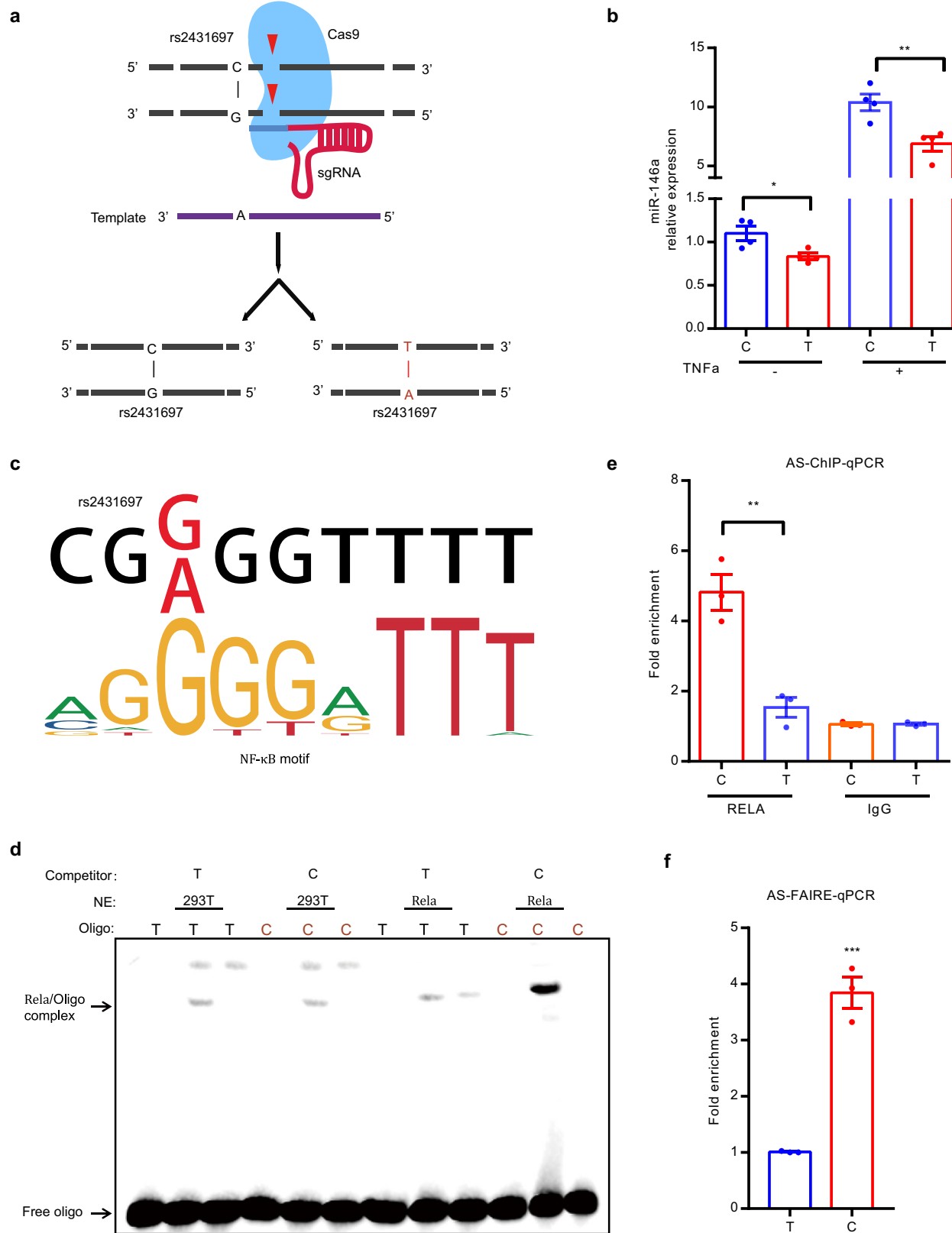

binding affinity, our results suggest a significant allele-specific chromatin state reflected by the H3K4me1 signal, H3K27ac signal, and chromatin accessibility, further detailing the mechanisms through which the rs2431697 T risk allele mediates its effects on SLE pathogenesis.

Prior to this study, miR-146a was considered to be a valid target for SLE intervention[32,33], in part because it targets several SLE risk genes[32,63] (Supplementary Data 6). Here we found that targeting the CRISPRa system to the rs2431697-containing region can effectively decrease the IFN score by upregulating miR-146a

**Fig. 5 The SLE risk-associated SNP rs2431697 alters NF-κB binding and the chromatin state to modulate miR-146a expression. a** Generation of homozygous clones harboring the major and minor alleles with CRISPR/Cas9 in U-937 cells. **b** RT-qPCR analysis demonstrates decreased miR-146a expression in T/T clones compared to C/C clones both at native and stimulatory conditions (*$P = 0.0239$, **$P = 0.0095$) (four biological samples replicates and three biological replicates). **c** NF-κB preferentially binds to the C non-risk allele of rs2431697, as predicted by bioinformatics analysis. **d, e** NF-κB favors binding to the C non-risk allele at rs2431697 as determined by EMSA (**d**) and ChIP followed by AS-qPCR (**e**) in rs2431697 heterozygous U-937 cell clones, **$P = 0.0021$ ($n = 3$, biological replicates). NE, nuclear extract. AS-ChIP-qPCR, allele-specific ChIP-qPCR. **f** The rs2431697 C allele has higher chromatin accessibility than the T allele. FAIRE signal is significantly higher at the rs2431697 region for the C allele compared to the T allele as examined by FAIRE followed by AS-qPCR, suggesting rs2431697 may alter chromatin accessibility of this locus, ***$P = 0.0005$ ($n = 3$, biological replicates). AS-FAIRE-qPCR, allele-specific FAIRE-qPCR. Data are represented as mean ± SEM, and $P$-values are calculated using unpaired two-tailed Student's $t$-test. *$P < 0.05$; **$P < 0.01$; ***$P < 0.001$. See also Supplementary Fig. 10.

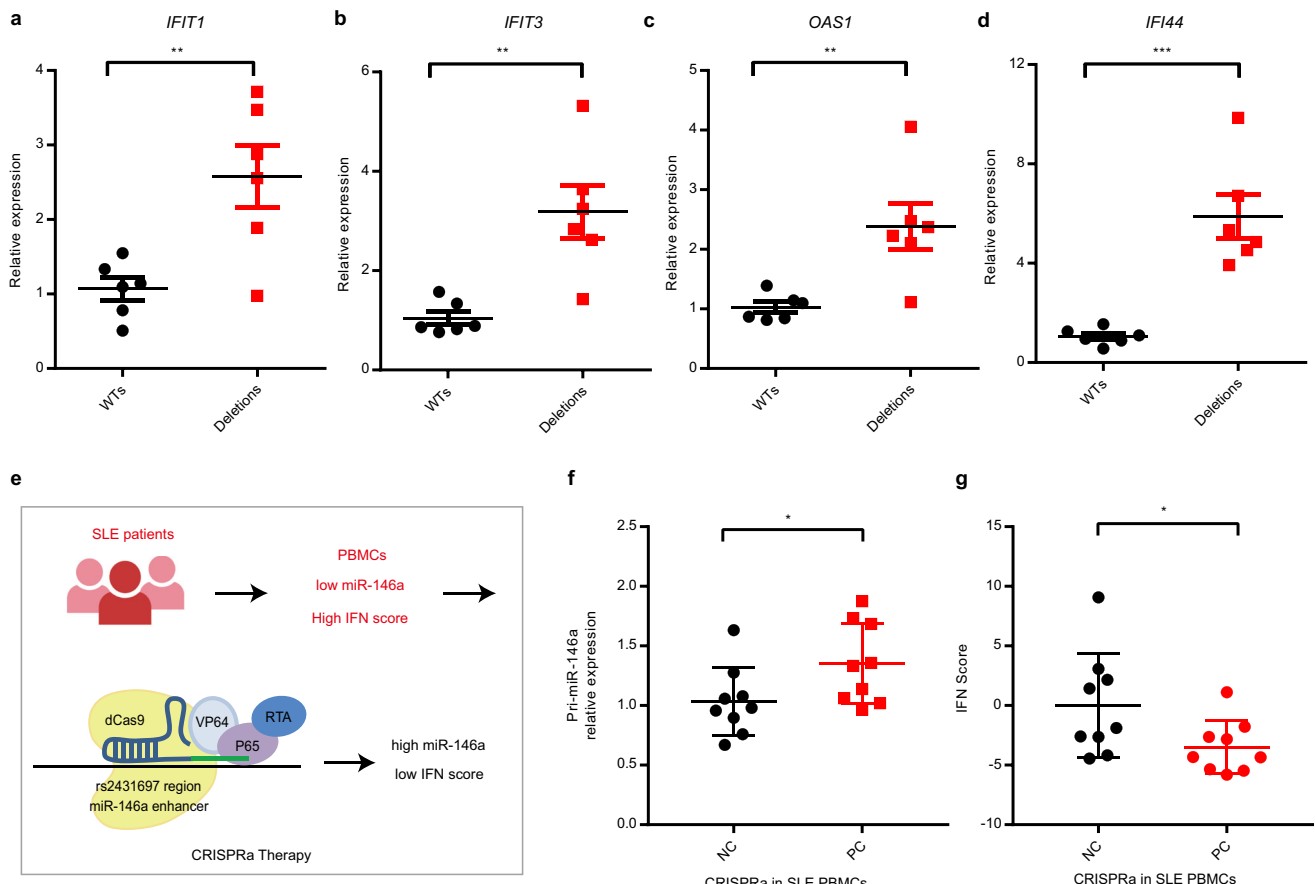

**Fig. 6 Targeting rs2431697 intervenes with the activation of the interferon pathway by modulating miR-146a expression. a–d** RT-qPCR analysis of interferon-inducible gene expression in U-937 WT and deletion clones after treating with Interferon-α (*IFIT1*: **$P = 0.0067$, **$IFIT3$: $P = 0.0027$, **$OAS1$: $P = 0.0066$, ***$IFI44$: $P = 0.0003$) ($n = 6$, replicates represent unique biological clone replicates). **e** Flow scheme of inhibiting SLE IFN pathway by CRISPR activation (CRIPSRa) approach. **f** RT-qPCR analysis of *Pri-miR-146a* expression in SLE patients' PBMCs after treating with the CRISPRa system targeting the rs2431697 locus, *$P = 0.0204$ ($n = 9$, replicates represent biological samples from unique individuals). **g** CRISPRa therapy decreases the IFN score of SLE patients' PBMCs based on sgRNAs targeting the rs2431697 site, *$P = 0.0158$ ($n = 9$, replicates represent biological samples from unique individuals). Data are represented as mean ± SEM, and $P$-values are calculated using unpaired two-tailed Student's $t$-test (**a–d**) and paired two-tailed Student's test (**f, g**). *$P < 0.05$; **$P < 0.01$; ***$P < 0.001$.

expression in SLE patients' PBMCs, suggesting that therapeutic manipulation of the enhancer region harboring rs2431697 could be applied to alleviate SLE development. Our data suggest that employing a CRISPR-based approach to suppress the development of SLE by targeting disease-related miRNAs is likely feasible. Therefore, our study provides a new strategy to treat human diseases by targeting the disease-associated regulatory elements of miRNAs.

In conclusion, our study illustrates a feasible strategy to localize ncRNA regulatory elements by exploiting the information provided by disease-risk SNPs. Our work elucidates the functional mechanism of risk variant rs2431697 in SLE pathogenesis and provides a novel insight for using genetic data to develop novel interventions.

## Methods

**Discovery cohort genetic analysis.** *Genotyping.* In the Discovery cohort, 97 SNPs were genotyped covering the *PTTG1-MIR3142HG* region (Supplementary Data 1), spanning GRCh37/hg19 chr5: 158879978–160879978, as part of a larger collaborative study, the LLAS2. Samples were collected from individuals in the United

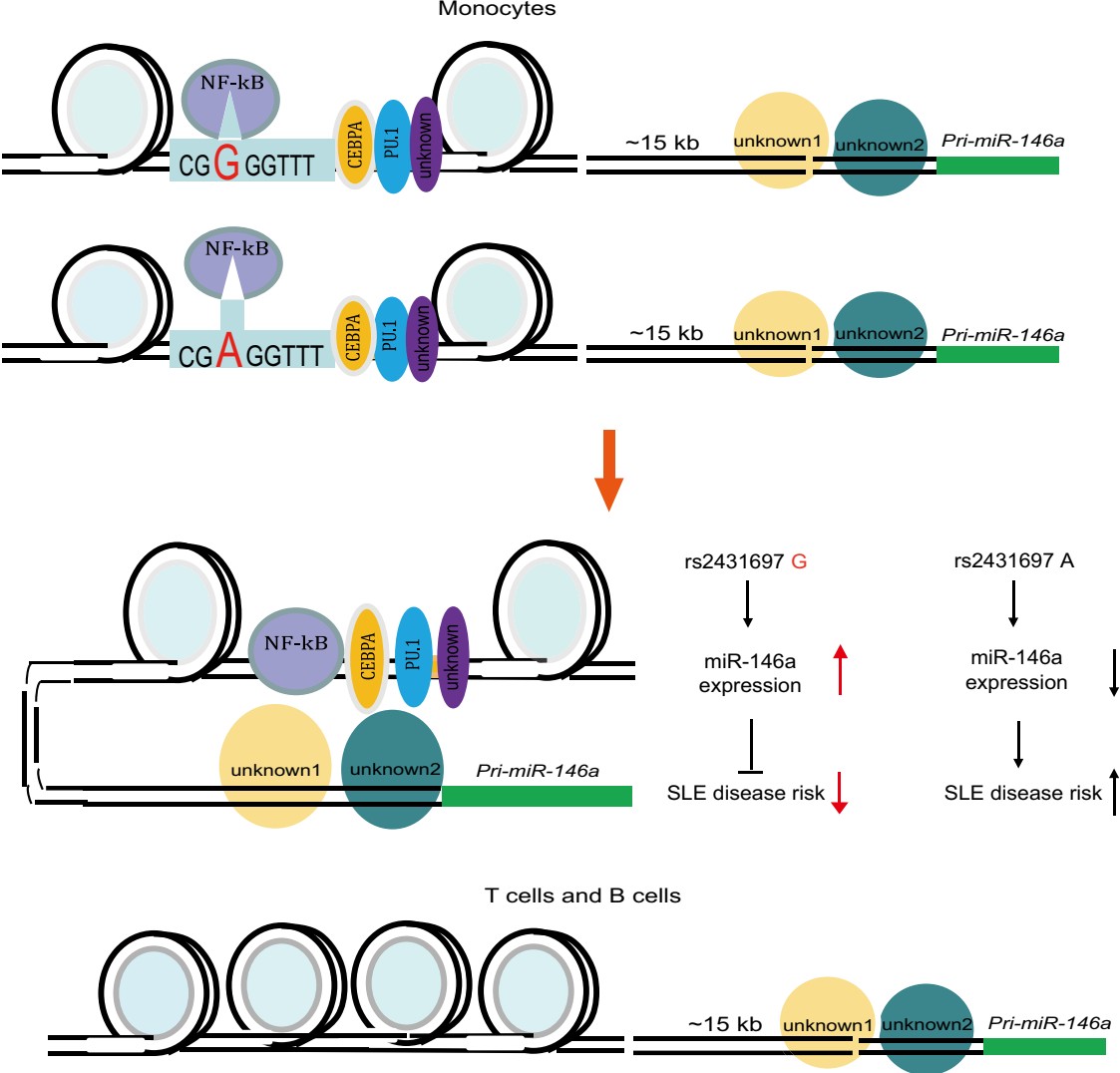

**Fig. 7 A model of rs2431697 regulating miR-146a expression in a cell-type-dependent manner.** Lineage-determining transcription factor PU.1 or other transcription factors shape the cell-specific enhancer at the rs2431697 locus and NF-κB acts as a stimulating transcription factor by binding to the genomic sequence harboring rs2431697. The rs2431697 A risk allele has lower NF-κB-binding affinity than the non-risk G allele, resulting in the reduction of miR-146a expression and increased SLE risk.

States, Asia, Europe, and Latin America. Genotyping was performed on the Illumina iSelect platform located at the Lupus Genetics Studies Unit at the Oklahoma Medical Research Foundation (OMRF). Subjects were grouped into four ancestral groups: EU, AA, AS, and AI. All cases met the American College of Rheumatology criteria for the classification of SLE[64] and both cases and controls were enrolled in this study through an informed consent process approved through the local Institutional Review Boards (IRBs) as detailed in (https://pubmed.ncbi.nlm.nih.gov/21194677/). Specifically, IRBs at Seattle Children's Hospital and University of Washington, Wake Forest University School of Medicine, University of California (Los Angeles), University of Southern California School of Medicine, University of Alabama at Birmingham, Medical University of South Carolina, OMRF, CIB Rosario University, University of California (San Francisco) and Feinstein Institute of Medical Research, University of Oklahoma Health Sciences Center, University of Colorado School of Medicine, Hanyang University Hospital for Rheumatic Diseases, Mayo Clinic, and King's College London provided recruitment oversight. LLAS2 included genotyping of other SLE risk loci and the analyses of those loci from this same collection, with and without SLE, have been published separately[65–79].

**Quality control and sample overlap**. Intensity data were obtained using standard procedures for all samples[79]. Briefly, SNPs were genotyped with Infinium chemistry on an Illumina iSelect custom array as per the manufacture's protocol. Only well-defined clusters for genotype calling were included for subsequent quality control filters. Samples with genotyping rates >95% for variants passing other

variant quality control filters were excluded. Variants with a per-group minor allele frequency <1% and per-variant call rate of <90% were excluded. Duplicated samples and first-degree relatives (πhat > 0.4) were excluded and the sample with the highest call rate retained. To ensure lack of overlap between the discovery and replication cohorts, samples from contributing individuals who were also authors on the publications in the replication cohort were removed. To ensure that samples in the European and European Ancestry discovery, and replication cohorts did not overlap, 3693 samples contributed by either Timothy Vyse, Marta Alarcon-Riquelme, and Lindsey Criswell were excluded from the "Discovery analysis" as all contributed samples to the study by Bentham et al.[41] Similarly, to ensure that samples in the replication cohort did not overlap with the discovery cohort, the summary statistics from the Korean ancestry group from the study by Sun et al.[40] (https://static-content.springer.com/esm/art%3A10.1038%2Fng.3496/MediaObjects/41588_2016_BFng3496_MOESM228_ESM.xlsx) were excluded, as Sang-Cheol Bae, a co-author on that study, contributed samples to the Discovery Cohort.

**Imputation to 1000 Genomes reference panel**. Genotyped data files were converted to Variant Call Format files. These were checked using the 1000 Genomes imputation preparation and checking scripts Version 4.2.11, developed by Will Rayner. In brief, these scripts harmonize the variants from the data files to be imputed to the reference panel, removing potentially ambiguous variants and those not in the reference panel. The harmonized.vcf files produced by these scripts were then uploaded to the Michigan Imputation Server. Imputation with minimac4 was

performed to 1000 Genomes Phase 3 v5 (GRCh37/hg19) using rsq Filter 0.3, Eagle v2.4, to the 1000 Genomes Mixed population and Quality Control and Imputation mode[38]. Prior to association analysis, the markers in the region with $r^2 < 0.8$ were excluded, as this cutoff has been shown to consistently yield reliable imputation data[80–82].

**Ascertainment of and correction for population stratification**. Subjects were self-identified in each ancestral population. Genetic outliers from each ancestral population group were removed from further analysis according to principal components and admixture estimates. Three of the four admixture estimates proportions were used as covariates in the statistical models in concert with prior analysis pipelines applied to this data set[79].

**Frequentist statistical analysis: discovery cohort**. Association with SLE at each variant was assessed using logistic regression models with three admixture estimates as covariates for the additive genetic model as implemented in PLINK v 1.9[83]. Analysis was carried out first on variants in the genotyped data set and subsequently on variants in the imputed data set. Following this, *trans*-ancestral meta-analysis combining the four ancestral populations in the discovery cohort was performed using METAL according to the SAMPLESIZE analysis scheme[84]. After identifying the rs2431697 as the variant most robustly associated with SLE, step-wise conditional logistic regression was performed using the rs2431697 genotype as a model covariate in the initial step and markers with residual evidence of association ($1E − 2 > P > 1E − 4$) in subsequent steps.

**Bayesian statistical analysis: discovery cohort**. Bayes factors (BFs) were calculated relating genotype configuration at each variant to disease status using SNPTEST v 2.5.2[85] and including three admixture estimates as covariates in the additive model. To identify the variants most likely responsible for the statistical association, we calculated posterior probability under the assumption that any variant responsible for a genetic effect could be causal, and that only one variant is causal for each genetic effect. The posterior probability for $SNP_i$ is the ratio between the BFs for $SNP_i$ and the sum of the $k$ SNPs in the region: $PP_i = \frac{BF_i}{\sum_{j=1}^{k} BF_j}$.

The 95% credible set was defined by including the minimum number of variants, where the sum total posterior probability in the set was ≥0.95.

**Summary statistic meta-analysis: replication cohort**. Summary statistics from three SLE association studies[40–42] were used in meta-analysis as a replication cohort. The summary statistics from the Bentham et al.[41] study were downloaded from immunobase.org (https://www.immunobase.org/downloads/protected_data/). The summary statistics from the Julia et al.[42] (http://urr.cat/data/GWAS_SLE_summaryStats.zip) and Sun et al.[40] (https://static-content.springer.com/esm/art%3A10.1038%2Fng.3496/MediaObjects/41588_2016_BFng3496_MOESM228_ESM.xlsx) studies were downloaded from hyperlinks located at the respective journal websites. The *P*-values were combined together using METAL according to the SAMPLESIZE analysis scheme[84]. As described in the Quality Control & Sample Overlap section, given the potential for overlap, the Korean ancestry samples from the Sun et al.[40] were excluded from the "Replication cohort" meta-analysis. Posterior probabilities and 95% credible sets for this and for the Discovery *trans*-ancestral meta-analysis were calculated based on *P*-values using the approach in LocusZoom.

**eQTL analysis**. RNA-Seq data[43] were downloaded from the eQTL resources from the Gilad/Pritchard group website. This gene expression data was collected 69 LCLs generated as part of the International HapMap project. The cell lines were derived from primary cells of YRIs from Ibadan, Nigeria. Genotype data in the region around rs2431697 from the YRI population from the 1000 Genomes Project was downloaded from https://grch37.ensembl.org/Homo_sapiens/Tools/DataSlicer?db=core. For the 58 sample IDs in this population common to both the 1000 Genomes Project and this RNA-Seq data set, individual variant genotypes were used as the predictor and normalized, pc-corrected gene expression values were used as the response variable in an additive linear model as implemented in PLINK v1.9[83]. The ensmbl gene identifiers from this data set were mapped to HGNC IDs using biomaRt _2.38.0[86]. Notably, *C1QTNF2* did not map to an ensemble gene identifier and its stable ensemble gene identifier was not found in the RNA-Seq data set, suggesting that this gene was not robustly expressed in these cell lines. A search for *trans*-eQTL association with rs2431697 was performed using each of the 22,032 mapped transcripts from this data set as the response variable in a similar manner.

**Cell lines and culture**. All cell lines were obtained from Cell Bank of Chinese Academy of Science (Shanghai, China). U-937, Raji, and Jurkat cells were maintained in RPMI-1640 media (11875-093, Gibco) supplemented with 10% fetal bovine serum (FBS) and 1% penicillin–streptomycin (15140122, Gibco). HEK-293T cells were maintained in DMEM High Glucose media (11965-092, Gibco) supplemented with 10% FBS (10099141, Gibco) and 1% penicillin–streptomycin. All cells were cultured at 37 °C and 5% $CO_2$. The cells were checked regularly for

mycoplasma MycoAlert Mycoplasma Detection Kit (LT07-118, Lonza). These cell lines were free of mycoplasma during our study.

**Genome editing in cell lines**. sgRNAs were designed using online CRISPR Design tool (crispr.mit.edu) and we selected the possible sgRNAs targeting the core DNA sequence of rs2431697 or miR-146A promoter based on high efficacy score and low potential off-target sites. For the genome-editing experiments in cell lines, a bicistronic vector PX458 (Addgene 48138), expressing a chimeric guide RNA, a human codon-optimized Cas9, and a GFP, was linearized by BbsI (R3059L, NEB), dephosphorylated, and then gel-purified. Guide RNA oligos were synthesized in Genewiz (Suzhou, China), annealed and subcloned into the linearized PX458 plasmid, plasmids were transformed into chemically competent *Escherichia coli* (Sbtl3, Transgen Biotech) and grown, and plasmid DNA was extracted and purified.

Cell lines were genotyped for rs2431697 via Sanger sequencing using a locus-specific primer. Cells were routinely cultured in RPMI-1640 media with 10% FBS until ready for transfection. For homology directed repair (HDR), $1.5 × 10^6$ cells were electroporated with 6 µg of CRISPR plasmid and 30 µg of a 121 bp single-stranded oligodeoxynucleotide donor template using 100 µL Neon Transfection System (MPK10025, Thermo Fisher). After transfection for 12 h, cells were treated with 1 µM SCR-7 (SML1546, Sigma-Aldrich) to enhance HDR efficiency. After transfection for 3 days, single cells with high GFP fluorescence were sorted into 96-well plates supplemented with 200 µL culture medium in each well. Following 14 days of cell growth, genomic DNA of individual colonies was isolated using TransDirect Animal Tissue PCR Kit (AD201-02, Transgen Biotech), target sequence was amplified by locus-specific primer, and sequenced by Sanger sequence to screen for desired mutations.

To delete the target noncoding sequence around rs2431697 and miR-146a promoter, we utilized a dual-guide RNA strategy using two Cas9-guide RNA constructs. Genomic deletions were screened with Sanger sequencing of PCR amplicons. Electroporation conditions for each cell line are as follows: U-937, 1400 v, 10 ms, 3 pulses; Raji, 1350 v, 30 ms, 1 pulse; Jurkat, 1350 v, 10 ms, 3 pulses.

**RNA extraction and quantitative RT-PCR analysis**. Total RNA was extracted using TRIzol reagent (15596018, Thermo Fisher). For mRNA analysis, cDNA was synthesized using PrimeScript™ RT Reagent Kit (Perfect Real Time) (RR037A, TAKARA) using 500 ng of RNA per cDNA reaction. Quantitative reverse transcriptipn PCR (RT-PCR) reactions were performed using TB Green Premix Ex Taq reagent (RR420A, TAKARA) according to the manufacturer's protocol and normalized by glyceraldehyde 3-phosphate dehydrogenase mRNA levels. For miRNA analysis, reverse transcription from miRNA to cDNA was performed using the TaqMan™ MicroRNA Reverse Transcription Kit (4366596, Thermo Fisher), using 200 ng of total RNA per reaction. miRNA expression was quantified by TaqMan MicroRNA Assays (4427975, Thermo Fisher) according to the manufacturer's protocol and normalized by RNU48 snRNA levels. Results were analyzed using the comparative Ct method normalizing to a control sample and housekeeping primers.

**PBMCs isolation**. Whole blood was collected from healthy human donors or SLE patient donors in sodium heparinized vacutainer tubes (Becton Dickinson, USA) with approval by the Committee on Human Research of Renji Hospital, Shanghai Jiao Tong University. All samples were collected with signed informed consent according to the Committee on Human Research of Renji Hospital. PBMCs were isolated by Ficoll gradient centrifugation. Fresh blood was mixed in a 1 : 2 ratio with phosphate-buffered saline (PBS) (Ca2+ and Mg2+ free) containing 2% FBS and 2 mM EDTA. Thirty-five milliliters of the respective PBS/blood solution was transferred to 50 mL Falcon tubes and underlaid with 15 mL Ficoll-Paque PLUS (17-1440-02, GE Healthcare). After density gradient centrifugation ($400 × g$, 35 min, no brakes), the PBMC layer was carefully removed and the cells were washed twice with PBS for further study.

**Lentivirus production**. HEK-293T cells were cultured as described above. One day before transfection, $3 × 10^5$ cells were seeded into the wells of the six-well plate. Cells were transfected on the next day. For each well, cells were transfected with 1 mg of plasmid containing the vector of interest, 250 ng of pMD2.G (Addgene 12259), and 750 ng of psPAX2 (Addgene 12260) using 3 µL of Lipofectamine 2000 (11668-019, Thermo Fisher). Six hours after transfection, the media was changed. Virus supernatant was collected at 72 h after transfection, centrifuged at 4 °C for 10 min, the supernatant was aliquoted, and stored at −80 °C.

**CRISPRi assay**. CRISPRi U-937 cell line stably expressing KRAB-dCas9-mCherry was generated using the pHR-SFFV-KRAB-dCas9-P2A-mCherry plasmid (Addgene 60954). Cells were transduced with lentivirus via spinfection in 12-well plates. Cells ($4 × 10^5$) in 500 µL of media supplemented with 10 µg/mL polybrene (107689, Sigma) were added to each well, supplemented with lentiviral supernatant, and centrifuged for 90 min at $1000 × g$. Twenty-four hours after spinfection, the media was changed and cells were cultured for another 48 h. Strong mCherry+ cells were sorted by FACS, and cultured for use in future experiments. sgRNAs targeting rs2431697-containing region were designed using online sgRNA design

tool chopchop (https://chopchop.cbu.uib.no/). Oligos were synthesized in Genewiz. Sequences were annealed and subsequently cloned into pKLV-U6-gRNA(BbsI)-PGKpuro2ABFP plasmid (Addgene 50946) using restriction enzyme BbsI (R3059L, NEB). sgRNA lentiviral particles were produced and stably expressing KRAB-dCas9-mCherry U-937 cells were transduced as mentioned above, the cells were further selected with 1 μg/mL puromycin (ant-pr-5, Invivogene) for 72 h.

**CRISPRa assay in U-937 cell line**. CRISPRa U-937 cell line stably expressing dCas9-VP64 and MS2-P65-HSF1 fusion proteins was generated using the lenti-dCAS-VP64-Blast plasmid (Addgene 61425) and lenti-MS2-P65-HSF1-Hygro (Addgene 61426) plasmid as mentioned above. Cells were selected with 10 μg/mL Blasticidin (ant-bl-5, Invivogene) and 300 μg/mL Hygromcin (10687010, Thermo Fisher) for 1 week. sgRNAs targeting rs2431697-containing region were cloned into lenti-sgRNA(MS2)-zeo backbone plasmid (Addgene 61427) using restriction enzyme BsmBI (R0580L, NEB); sgRNA lentivirus particles were produced and transduced as mentioned above. Cells were selected with 400 μg/mL Zeocin (R25001, Thermo Fisher) for 72 h.

**CRISPRa assay in PBMCs of SLE patients**. For CRISPRa assay in SLE PBMCs, we first amplified the U6 promoter-sgRNA-sgRNA scaffold(MS2) fragment from 61,427 plasmid expressing different CRISPRa sgRNAs and subcloned into the 61,426 plasmid using BmtI restriction enzyme site with Gibson Assembly method. This constructed plasmid expresses a chimeric guide RNA with modified sgRNA scaffold containing MS2 stem loop, MS2-N55K-p65-HSF1 fusion protein, and we just need to transfect two plasmids instead of the three-plasmid system in the CRISPRa assay. PBMCs were isolated from SLE patients with high IFN score (high expression of interferon-inducible genes); $3.5 \times 10^6$ PBMCs from SLE patients were transfected with 10 μg 61425 plasmid, 10 μg constructed plasmid expressing sgNC (MS2)-MS2-P65-HSF1, or 10 μg constructed plasmid expressing sg2a(MS2)-MS2-P65-HSF1 by Neon transfection system; the transfection condition is 2150 V, 20 ms, 1 pulse. After transfection for 24 h, the cells were collected and RNA was extracted to test miR-146a and interferon-inducible gene expression.

**crRNA–tracrRNA duplex preparation**. Alt-R crRNAs and Alt-tracrRNA-ATTO550 (1075928, IDT) were purchased from IDT and reconstituted to 200 μM with Nuclease-Free Duplex Buffer (IDT). Two RNA oligos were mixed in equimolar concentrations in a sterile microcentrifuge tube to a final duplex concentration of 44 μM. Oligos were annealed by heating at 95 °C for 5 min and slowly cooled to room temperature.

**Cas9 RNP assembly**. For each reaction, 22 pmol crRNA–tracrRNA duplex and 18 pmol HiFi Cas9 Nuclease (1081061, IDT) were combined in Buffer T to a final volume 1 μL, gently mixed and incubated at room temperature for 10 min.

**Human primary cells isolation, culture, and editing**. Human CD3+ T cells, CD14+ monocytes, and CD19+ B cells were separately isolated from PBMCs using the Human CD3+ T Cell Isolation kit (130-050-101, Miltenyi Biotec), Human CD14+ monocytes Isolation kit (130-050-201, Miltenyi Biotec), and Human CD19+ B Cell Isolation kit (130-050-301, Miltenyi Biotec) according to the manufacturer's protocol. Purities were >95%. T cells were cultured in OpTmizer™ CTS™ T-Cell Expansion SFM medium (A10458-03, Thermo Fisher) with CD3/CD28 dynabeads (11131D, Thermo Fisher) and 1% penicillin–streptomycin. Monocytes were cultured in RPMI-1640 media with 10% (vol/vol) HI-FBS, 2 mM L-Glutamine (25030081, Thermo Fisher), 55 μM β-mercaptoethanol (M6250, Sigma-Aldrich), and 1% penicillin–streptomycin. B cells were cultured in RPMI-1640 media with 10% (vol/vol) HI-FBS, 2 mM L-Glutamine, 55 μM β-mercaptoethanol, and 1% penicillin–streptomycin, 50 IU/mL IL-4 (200-04, Peprotech), and crosslinked CD40 ligand (130-098-775, Miltenyi Biotec).

For T-cell transfection with Cas9 RNP, T cells were first activated with CD3/CD28 dynabeads in OpTmizer™ CTS™ T-Cell Expansion SFM medium for 48 h; 6 h before transfection, CD3/CD28 dynabeads were removed. Cells ($2 \times 10^5$) were washed twice with PBS and resuspended into 9 μL of Buffer T, mixed with Cas9 RNP, and electroporated using the Neon transfection system with the condition 1400 V, 10 ms, 3 pulses. After electroporation, T cells were transferred to 500 μL of their respective culture medium supplemented with 30 IU/mL IL-2 (200-02A, Peprotech) but without CD3/CD28 dynabeads in a 48-well plate. Three days after electroporation, miR-146a expression was detected by Taqman methods and editing efficiency was analyzed by T7 endonuclease I assay.

For B-cell transfection with Cas9 RNP, B cells were first cultured in the respective medium for 48 h, $1.2 \times 10^5$ cells were washed twice with PBS and resuspended into 9 μL of Buffer T, mixed with Cas9 RNP, and electroporated using the Neon transfection system with the condition 1400 V, 10 ms, 3 pulses. Cells were transferred to 500 μL of their respective culture medium in a 48-well plate. Three days after electroporation, miR-146a expression was detected by Taqman methods and editing efficiency was analyzed by T7 endonuclease I assay

For monocyte transfection with Cas9 RNP, after isolation, $2.5 \times 10^5$ monocytes were directly washed with PBS and resuspended into 9 μl of Buffer T, mixed with Cas9 RNP, 1 μL Alt-R Cas9 Electroporation Enhancer, and electroporated using the Neon transfection system with the condition 1600 V, 10 ms, 3 pulses. Cells were

transferred to 200 μL of their respective culture medium in a 96-well plate. One day after electroporation, miR-146 expression was detected by Taqman methods and editing efficiency was analyzed by T7 endonuclease I assay.

**DNA extraction from human primary cells**. DNA of human primary cells was isolated using TRIzol™ Reagent. Briefly, cells were lysed with TRIzol™ Reagent, chloroform was added, mixed, and centrifuged. The aqueous phase was removed to extract RNA, the DNA in interphase was precipitated by 100% ethanol, and washed with 0.1 M sodium citrate and 75% ethanol to get DNA.

**Detection of editing efficiency in human primary cells by T7 Endonuclease I Assay**. PCR amplicons spanning the sgRNA genomic target sites were generated using the $2 \times$ High-Fidelity Master Mix (TP001, Tsingke) with the locus-specific primer. PCR amplicons were purified and 250 ng was denatured and re-annealed in a thermocycler and digested with T7 Endonuclease I (M0302L, NEB) according to the manufacturer's protocol. Digested DNA was run on a 2% agarose gel containing GoldView II Nuclear Staining Dyes (G8142, Solarbio) and visualized on a ChemiDoc XRS+ (Bio-Rad). Band intensities were analyzed using Image Lab Software (ImageJ 1.51). The percentage of editing was calculated using the following equation $[1 - (1 - (b + c/a + b + c))^{1/2}] \times 100$[87], where $a$ is the band intensity of DNA substrate, and $b$ and $c$ are the cleavage products.

**Mice and engraftment of human PBMCs**. All animal studies were approved by the Animal Care Committee of Renji Hospital. NSG mice were purchased from The Jackson Laboratory. The mice were kept under specific pathogen-free conditions in individually ventilated cages under a 12 h light/12 h dark cycle in a 23 ± 2 °C temperature and 50 ± 10% humidity. Six- to 8-week-old female mice were used for engraftment of human PBMC. PBMCs were isolated by standard Ficoll separation, washed in PBS solution, and suspended in RPMI-1640. PBMCs ($1 \times 10^7$) were injected into the mice by tail vein injection.

**Adenovirus-Cas9-sgRNAs mediated genome editing in vivo**. Adenovirus expressing Cas9-GFP and dual sgRNAs targeting rs2431697 region or negative control dual sgRNAs were produced and purified in Hanbio company (Shanghai, China). At 24 days post-inoculation of human PBMCs, 12 mice were randomly assigned to two groups. Mice were treated with $2 \times 10^{10}$ plaque forming unit (PFU) negative control adenovirus or targeting rs2431697 region adenovirus. After 3 days of transduction in vivo, the PBMCs and total spleen cells were isolated. The cells were stained directly with cocktails of fluorescently conjugated antibodies (APC-H7 Mouse anti-Human CD45 (560178, BD Biosciences, 1:50), BV421 Mouse Anti-Human CD19 (562440, BD Biosciences, 1:50), PE-Cy™7 Mouse Anti-Human CD14 (562698, BD Biosciences, 1:50), APC Mouse Anti-Human CD3 (555335, BD Biosciences, 1:25)) and sorted on a BD FACSAria III (Becton Dickinson) using the gating strategies shown in Supplementary Fig. 9d; flow cytometry data were analyzed using FlowJo software (version 10). For each immune cell type, 100 GFP+ cells were sorted to perform the quantitative RT-PCR analysis.

**miRNA expression analysis of in vivo editing immune cells**. This assay was performed with miScript Single Cell qPCR Kit (331055, Qiagen) according to the manufacturer's instructions. Briefly, 100 GFP+ human CD14+ cells or CD19+ cells, or CD3+ cells were sorted from humanized mice. After cell lysis, 3′ and 5′ adapters are ligated to mature miRNAs. The ligated miRNAs were then reverse-transcribed to cDNA. Following cleanup, the cDNA was pre-amplified and used for real-time PCR expression analysis.

**Chromatin immunoprecipitation-qPCR**. ChIP assay was performed with SimpleChIP® Plus Enzymatic Chromatin IP Kit (9005, Cell Signaling Technology) according to the manufacturer's instructions. Briefly, $1 \times 10^7$ cells per ChIP assay were crosslinked with 1% formaldehyde solution and quenched with 0.125 M glycine. Cells were washed with PBS and centrifuged for 5 min at $210 \times g$. Cell pellets were resuspended in 2 mL of 1× Buffer A buffer for 10 min at 4 °C, centrifuged, and followed by 2 mL of 1× Buffer B buffer, and finally resuspended in 200 μL 1× Buffer B buffer. For per IP reaction, 1 μL of Micrococcal Nuclease was added, mixed by inverting tube several times and incubate for 30 min at 37 °C to digest DNA. The reaction was stopped by adding 20 μL of 0.5 M EDTA and pellet nuclei was collected by centrifugation at $15,800 \times g$ for 1 min at 4 °C. Nuclear pellet was finally resuspended in 100 μL of 1× ChIP Buffer, incubate on ice for 10 min, and subsequently sonicated at 4 °C with a Bioruptor sonicator (Diagenode) at high power for 5 cycles for 30 s with 30 s between cycles. Supernatant was collected and incubated with antibodies (RELA (8242S, Cell Signaling Technology, 1 : 100), H3K4me1 (ab8895, Abcam, 2 μg for 25 μg of chromatin), and H3K27ac (ab177178, Abcam, 2 μg for 25 μg of chromatin)) bound to Magna ChIP™ Protein A + G Magnetic Beads (16-663, Millipore) at 4 °C overnight. Beads were washed three times with low-salt wash buffer and once with high-salt wash buffer. Cross-links were reversed overnight. RNA and protein were digested using RNase A and Proteinase K, respectively. DNA fragments were purified using Spin Columns in this kit. For ChIP-qPCR analysis, the enrichment of target DNA fragments was

calculated using the comparative Ct method normalizing to IgG negative control sample..

**Formaldehyde-assisted isolation of regulatory element-qPCR**. FAIRE analysis used an aliquot from crosslinked and sonicated ChIP chromatin to prepare the FAIRE DNA sample. The chromatin lysate was extracted twice with phenol/chloroform/isoamyl alcohol and then with chloroform/isoamyl alcohol. FAIRE DNA in the aqueous phase was precipitated with ethanol in the presence of glycogen and resuspended in 10 mM Tris-HCl (pH 7.4). FAIRE DNA and control underwent reversal of cross-links, purification and were analyzed by quantitative RT-PCR with specific primers targeting DNA sequences at different distances to rs2431697 site. Values were normalized to input DNA and compared to a region just outside the putative regulatory region.

**Allele-specific qPCR**. AS-qPCR primers were designed to specifically amplify the rs2431697 region with a T or C allele in the DNA samples from ChIP or FAIRE assays. AS-qPCR was assayed similarly to normal qPCR.

**DAPA mass spectrometry**. Cells ($1 \times 10^7$) were first lysed with cytoplasmic extraction buffer with a final concentration of 10 mM HEPES pH 7.9, 10 mM KCl, and 0.1 mM EDTA, then were further lysed with nuclear extraction (NE) buffer with a final concentration of 20 mM HEPES pH 7.9, 0.4 M NaCl, and 1 mM EDTA to get nuclear extract[88]. Protein concentrations of nuclear lysates were detected by Bradford's method. Nuclear extracts (200 μg) were mixed with 50 pmol of specific or nonspecific 5′-biotinylated DNA probes in 400 μL of Buffer D (20 mM HEPES pH 7.9, 10% glycerol, 50 mM KCl, 0.2 mM EDTA, 1.5 mM MgCl₂, 100 μg/mL Sheared Salmon sperm DNA, 1 mM dithiothreitol, and 0.25% Triton X-100) supplemented with protease inhibitor cocktail and incubated on ice for 45 min. Fifty microliters of Dynabeads™ M-280 Streptavidin (11205D, Thermo Fisher) was then added to each reaction and rotated for 2 h at 4 °C. The beads-DNA probe-proteins complexes were subjected to three washes with 500 μL Buffer D. Proteins were dissociated from the beads by the addition of 2 x Laemmli sample buffer (161-0737, Bio-Rad) and boiled at 95 °C for 10 min. The boiled protein samples underwent SDS-polyacrylamide gel electrophoresis and the gels were stained with Coomassie dye. The whole lane was cut into pieces and digested by trypsin for MS analysis.

**Liquid chromatography-MS/MS analysis and data processing**. The tryptic peptides were dissolved in 0.1% formic acid (solvent A), directly loaded onto a home-made reversed-phase analytical column (15 cm length, 75 μm i.d.). The gradient comprised an increase from 6% to 23% solvent B (0.1% formic acid in 98% acetonitrile) over 16 min, 23% to 35% in 8 min and climbing to 80% in 3 min then holding at 80% for the last 3 min, all at a constant flow rate of 400 nl/min on an EASY-nLC 1000 UPLC system.

The peptides of two control groups and two experiment groups were subjected to NSI source followed by tandem MS (MS/MS) in Q ExactiveTM Plus (Thermo) coupled online to the UPLC. The electrospray voltage applied was 2.0 kV. The m/z scan range was 350–1800 for full scan and intact peptides were detected in the Orbitrap at a resolution of 70,000. Peptides were then selected for MS/MS using NCE setting as 28 and the fragments were detected in the Orbitrap at a resolution of 17,500. A data-dependent procedure that alternated between one MS scan followed by 20 MS/MS scans with 15.0 s dynamic exclusion. Automatic gain control was set at 5E4.

The resulting MS/MS data were processed using Mascot Daemon (version 2.3.0). Tandem mass spectra were searched against 2019-uniprot-human database. Trypsin/P was specified as cleavage enzyme allowing up to two missing cleavages. Mass error was set to 10 p.p.m. for precursor ions and 0.02 Da for fragment ions. Carbamidomethyl on Cys were specified as fixed modification and oxidation on Met was specified as variable modification. Peptide confidence was set at high and peptide ion score was set >20. The candidate binding proteins must meet the following criteria: (1) proteins must be detected only in experiment group; (2) proteins must be detected in two replicate studies; (3) proteins should be TFs based on our group found that TFs occupy multiple loci associated with complex genetic disorders[52].

**Electrophoretic mobility shift assay**. The gel-shift assay was carried out with the LightShift Chemiluminescent EMSA Kit (20148, Thermo Scientific) according to the manufacturer's instructions. Nuclear extracts of HEK-293T cells transiently transfected with plasmids expressing RELA or negative control cells were prepared as described in above DAPA-MS method. Protein concentrations of nuclear lysates were detected by Bradford's method. To prepare duplexes, sense and antisense oligonucleotides carrying either the T allele or C allele at rs2431697 spanning 40 bp around the SNP were combined and heated at 95 °C for 5 min followed by slow cooling to 25 °C. For competition experiments, binding reactions were first incubated with 1 pmol of competitor duplex at 25 °C for 10 min, 10 fmol of 5′-biotinylated oligonucleotide duplex were then added for further incubation at 25 °C for 15 min. Reactions were separated using 6% TBE gels in 0.5× TBE, electrophoretically transferred to Biodyne B nylon membranes (77016, Thermo

Scientific), crosslinked onto the membranes under 254 nm ultraviolet light for 10 min and detected by chemiluminescence.

**Circular chromosome conformation capture sequencing**. To perform 4 C experiment, $1 \times 10^7$ cells were crosslinked in 1% formaldehyde for 10 min at room temperature. After crosslinking, cells were quenched with 125 mM Glycine solution. Cells were collected and incubated in lysis buffer while tumbling for 10 min at 4 °C. After lysis, the pellet of nuclei was washed and resuspended in Csp6I buffer. SDS (0.3%) was added and incubated at 37 °C in a shaker at 750 r.p.m. for 1 h. Then, 2% Triton X-100 was added and cells were incubated for another hour to sequester the SDS. Csp6I enzyme (400 units; FD0214, ThermoFisher) (for promoter view point, Csp6I was replaced with DpnII (R0543M, NEB)) and were added for a 4 h incubation at 37 °C in a shaker at 900 r.p.m.; another 400 units of Csp6I were added for overnight incubation. SDS (1.5%) was added and incubated for 20 min at 65 °C to inactivate Csp6I. Cells were suspended with ligation buffer and 1% Triton X-100 to a volume of 7 mL and then incubated for 1 h at 37 °C with occasional shaking. 100 units of T4 DNA ligase (M0202S, NEB) were added and incubated for 4 h at 16 °C, followed by 30 min at 25 °C. After that, 300 mg Proteinase K was added for overnight incubation at 65 °C and 30 μl of RNaseA solution was added and incubated for 45 min at 37 °C. DNA was purified using phenol–chloroform and dissolved in restriction buffer with 50 units of NlaIII (R0125L, NEB)and incubated for overnight in a shaker at 37 °C and 900 rpm, followed by treatment with 65 °C for 25 min to heat inactivation the enzyme. The sample was diluted with ligation buffer to a final volume of 14 mL, 100 units of ligase were then added for incubation for 4 h at 16 °C, followed by 30 min at 25 °C. DNA was purified using phenol–chloroform and further purified with QIAquick PCR purification kit (28106, Qiagen). The DNA concentration was detected by Qubit (ThermoFisher). Each template (1 μg) was amplified using the Expand Long Template PCR kit (Roche, #11681834001) to construct the 4C-seq library with locus-specific primers containing Illumina sequences. The libraries were purified and sequencing was performed on a HiSeq × ten (Illumina). 4C-seq data were analyzed using the software pipeline[89–91], 4Cseqpipe (version 0.7), with settings: -stat_type median, -trend_resolution 2000. Normalized trend was computed within the genomic region (chr5: 159,770,001–159,980,000) for both viewpoints. Bowtie-align (version 1.2)[92] was used to map captured reads to the Homo Sapiens genome assembly GRCh37 (hg19) with the setting: -m 1 and captured fragments on chromosome 5 (reads per million more than 20) were listed. Circos plot were created with circus (v0.69-6). All 4C-seq experiments were performed with at two biological replicates.

**RNA-seq and gene expression analysis**. Total RNA of each sample was extracted using TRIzol Reagent. RNA quality was quantified and qualified by Agilent 2100 Bioanalyzer (Agilent Technologies). Total RNA (1 μg) with RNA integrity number (RIN) value above or equal 8 was used for following library preparation. Library was made using Illumina NEBNext® Ultra™ Directional RNA Library Prep Kit (E7420L, NEB) according to the manufacturer's protocol and the rRNA was depleted from total RNA using Ribo-Zero™ rRNA removal Kit. The libraries were loaded on an Illumina HiSeq X ten instrument accordingthe to manufacturer's instructions (Illumina). Sequencing was carried out using a $2 \times 150$ paired-end configuration, image analysis and base calling were conducted by the HiSeq Control Software (HCS) (v3.3.76.1) + OLB (v1.9.3) + GAPipeline-1.6 (Illumina) on the HiSeq instrument. Computation analysis of paired-end reads was conducted using cutadapt (v1.15), Samtools (v0.1.19), Hisat2 (v2.1.0), and HT-seq (v0.11.2) software. Statistical normalization and differential analysis were performed in R using DESeq2 (v1.24.0) package. The threshold to define up or down regulation was fold-change > 1.5 and FDR < 0.05. Visualization were also conducted in R (v3.3.3).

**ATAC-seq and analysis**. ATAC-seq was performed with the Illumina Nextera DNA Preparation kit (FC-121-1030, Illumina)[93]. Briefly, 50,000 live cells were freshly isolated and resuspended in lysis buffer (10 mM Tris-HCl pH 7.5, 10 mM NaCl₂, 3 mM MgCl₂, 0.1% NP40) and pelleted. The cell pellet was resuspended in transposase reaction mix (12.5 μl 2× TD buffer, 1.25 μL transposase, 11.25 μL nuclease-free water) and incubated at 37 °C for 30 min. Fragmented DNA was purified using Zymo DNA concentrator kit (D4014, Zymo Research) and library was generated by PCR using KAPA HiFi PCR master mix (KK2602, Kapa Biosystems). An appropriate number of PCR cycles was determined by qPCR PCR cleanup of libraries was performed using AMPure beads (A63881, Beckman Coulter) at a 1.2 ratio. Libraries were then sequenced on Illumina HiSeq × Ten with paired-end reads. Reads were mapped using Bowtie2 (v2.3.5) followed by removing PCR duplicates using Picard (v2.19.0). Peaks were called using MACS2 (v2.1.2).

**Calculation of IFN scores**. Four representative IFN-inducible genes (*IFI27*, *IFIT3*, *OAS1*, and *LY6E*) were chosen to calculate IFN scores[94,95]. Briefly, the mean expression level of *IFIT3* in the negative control group (sample was transfected with negative control sgRNAs) was calculated and subtracted from the expression level of *IFIT3* of each sample in negative control group and positive control group (samples were transfected with positive control sgRNAs), and the remainder was divided by the SD value of *IFIT3* in negative control group to standardize the gene

expression level. The standardized expression levels of *IFIT3*, *OAS1*, and *LY6E* were calculated in the same way. Finally, the four values were summed together to yield the reported IFN score for each sample. The mean IFN score of the CRISPRa-negative control was 0 (range, −4.45 to 9.08), and the mean IFN score in CRISPRa positive control was −3.48 (range, −5.47 to 1.12).

All oligoes used in this paper are listed in Supplementary Tables in Supplementary Information.

**Statistical analysis**. All statistical analyses were performed using R Studio (version 1.0.136) with R version 3.3.3 and GraphPad Prism 7 or 8 software. Data are shown as mean ± SEM. "*n*" represents the number of technical replicates of the representative biological replicate unless otherwise mentioned. Details of the statistical analysis for each experiment can be found in the relevant figure legends. All statistical analyses were calculated using paired or unpaired two-tailed Student's *t*-test as indicated in the the figure legend, unless otherwise mentioned. Asterisks define the significance level (*$P \le 0.05$; **$P \le 0.01$; ***$P \le 0.001$).

**Reporting summary**. Further information on research design is available in the Nature Research Reporting Summary linked to this article.

## Data availability

The RNA sequencing data, ATAC sequencing, and 4C sequencing data that support the findings of this study have been deposited in ArrayExpress database under accession codes E-MTAB-8978, E-MTAB-8982, and E-MTAB-9581, respectively. The mass spectrometry proteomics data that support the findings have been deposited to the ProteomeXchange Consortium via the PRIDE partner repository with the dataset identifier PXD021335. GWAS and Immunochip Summary statistics from three SLE association studies[40–42] were downloaded from immunobase.org (http://urr.cat/data/GWAS_SLE_summaryStats.zip, now also available at https://www.ebi.ac.uk/gwas/studies/GCST005831 and https://static-content.springer.com/esm/art%3A10.1038%2Fng.3496/MediaObjects/41588_2016_BFng3496_MOESM228_ESM.xlsx). The data from immunobase.org has since been re-located to https://www.ebi.ac.uk/gwas/studies/GCST003156. Genotype data for the eQTL analysis were accessed from the 1000 Genomes Project and was downloaded from https://grch37.ensembl.org/Homo_sapiens/Tools/DataSlicer?db=core. Gene expression data for the eQTL analysis were downloaded from http://eqtl.uchicago.edu/RNA_Seq_data/results/. All other remaining data are available within the article and Supplementary Files, or available from the authors upon request. All data are available from the corresponding author upon reasonable request. Source data are provided with this paper.

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

## Acknowledgements

This study was supported by grants from the National Natural Science Foundation of China (31630021, 31930037, and 81102266), National Human Genetic Resources Sharing Service Platform (2005DKA21300), Shanghai Municipal Key Medical Center Construction Project (2017ZZ01024-002), Shenzhen Science and Technology Project (JCYJ20180504170414637), Shenzhen Futian Public Welfare Scientific Research Project (FTWS2018005), and Sanming Project of Medicine in Shenzhen (SZSM201602087). I.T. W.H. received support from a NIAMS training grant (5T32AR007534-32), a resident preceptorship and a scientist development award from the Rheumatology Research Foundation.

## Author contributions

N.S. and G.J.H. designed the project. G.J.H., T.Z., and N.X. performed the experiments. Y.T.Q. and Y.O.Y. performed the ATAC-seq experiment. X.M.L. and L.C.K. performed the MPRA screen. T.Z., N.X., X.Y., and H.X. established the mouse model. J.Y.M., J.D., and D.D. helped sort the cells and analyzed the FACS data. I.T.W.H., J.B.H., Y.J.T., and B.N. analyzed the genetic association data. I.T.W.H. performed the eQTL analysis. H.H. D., Z.H.Y., and Z.Z.Y. collected the patients' samples. M.Z. and Y.G. analyzed the 4C-seq data. G.J.H., C.Y., X.Y.Z., and M.T.W. analyzed the bioinformatics data. N.S., G.J.H., and I.T.W.H. prepared the manuscript.

## Competing interests

The authors declare no competing interests.

## Additional information

[1]Shanghai Institute of Rheumatology, Renji Hospital, Shanghai Jiao Tong University School of Medicine (SJTUSM), Shanghai 200001, China. [2]State Key Laboratory of Oncogenes and Related Genes, Shanghai Cancer Institute, Renji Hospital, Shanghai Jiao Tong University School of Medicine (SJTUSM), Shanghai 200032, China. [3]Shanghai Institute of Rheumatology, China-Australia Centre for Personalized Immunology, Renji Hospital, Shanghai Jiao Tong University School of Medicine (SJTUSM), Shanghai 200001, China. [4]Shenzhen Futian Hospital for Rheumatic Diseases, Shenzhen 518040, China. [5]Division of Immunobiology, Cincinnati Children's Hospital Medical Center, Cincinnati, Ohio, 45229, USA. [6]Division of Rheumatology, School of Medicine, University of Colorado, Aurora, Colorado, 80045, USA. [7]Department of Immunology and Microbiology, School of Medicine, University of Colorado, Aurora, Colorado, 80045, USA. [8]Center for Autoimmune Genomics and Etiology, Cincinnati Children's Hospital Medical Center, Cincinnati, Ohio, 45229, USA. [9]Shanghai Institute of Nutrition and Health, Shanghai Institutes for Biological Sciences(SIBS), University of Chinese Academy of Sciences, Chinese Academy of Sciences (CAS), Shanghai 200031, China. [10]Department of Obstetrics and Gynecology, Renji Hospital, Shanghai Jiao Tong University School of Medicine (SJTUSM), Shanghai 200127, China. [11]Shanghai Key Laboratory of Gynecologic Oncology, Renji Hospital, Shanghai Jiao Tong University School of Medicine (SJTUSM), Shanghai 200127, China. [12]Sheng Yushou Center of Cell Biology and Immunology, Joint International Research Laboratory of Metabolic and Developmental Sciences, School of Life Sciences and Biotechnology, Shanghai Jiao Tong University (SJTU), Shanghai 200240, China. [13]Department of Pediatrics, University of Cincinnati College of Medicine, Cincinnati, Ohio, 45229, USA. [14]Division of Biomedical Informatics, Cincinnati Children's Hospital Medical Center, Cincinnati, Ohio, 45229, USA. [15]Division of Developmental Biology, Cincinnati Children's Hospital Medical Center, Cincinnati, Ohio, 45229, USA. [16]Division of Allergy and Immunology, Cincinnati Children's Hospital Medical Center, Cincinnati, Ohio, 45229, USA. [17]US Department of Veterans Affairs Medical Center, Cincinnati, Ohio, 45229, USA. [18]These authors contributed equally: Guojun Hou, Isaac T.W. Harley. ✉email: nanshensibs@gmail.com

