## [Peer Review File · Nature Communications]

REVIEWER COMMENTS

Reviewer #1 (Remarks to the Author):

Hou/Harley et al combine experimental and computational approaches to pinpoint the function of an SLE-associated non-coding variant to an enhancer controlling a microRNA expression. The microRNA in question, miRNA-146a, has a known role in inflammation and was already hypothesised (based on limited evidence) to underpin SLE associations in this locus. Nonetheless, the authors validate this association in an impressively comprehensive way, which probably makes their contribution worth seeing light in a high-impact journal such as Nat Comms.

Technical comments:

1. Lines 433-441:

Can the authors please show the actual quantitative data for 4C-seq beyond the CIRCOS plots.

2. Lines 462-471:

My understanding is that at present, mass spectrometry-based identification of factors binding to a given non-repetitive DNA sequence remains highly challenging, with potentially noisy results. Is this why the data to this extent is confined to a supplementary figure? I would like the authors to elaborate more on how this method was performed and its limitations.

3. Lines 497-504:

a) I am a little confused as to how the MARIO allele-specific analysis works. What exactly is the null model? Did I get it right that it's a non-parametric method?

b) Did the authors benchmark their findings against WASP and/or RASQUAL? This would be very welcome.

c) Could the authors elaborate why allele-specific signal was only formally detected in ChIP-seq data but not in ATAC-seq and FAIRE-seq?

4. Could the authors elaborate on the direct targets of miRNA-146a? Are any of them already known and validated? If not, could the authors at least run some target prediction algorithms and report the results?

Minor textual comments:

Line 479: Suggest mentioning explicitly that RELA is an NFkB subunit.

Lines 510-515: This sentence is a bit too long and convoluted. Suggest splitting it into two or more to take the reader through the rationale for this experiment a bit more slowly and clearly.

Reviewer #2 (Remarks to the Author):

This study combines genomic, genetic, epigenetics and mechanistic assays to show that rs2431697 is a causal SNP that regulates miR-146a expression with the T/A allele being protective in a cell-specific manner. The authors show convincingly that this SNP corresponds to an open chromatin, epigenetically marked enhancer region of miR-146a in monocytes, and that the risk allele binds NF-kB more than the protective allele. Moreover, the last experiment shows that the deletion of the SNP region in monocytes from SLE patients lowers their high IFN score, providing a nice proof of principle of its causal effect in its association with SLE susceptibility. This study provides a definitive mechanism

linking low miR-146a expression in SLE susceptibility and rs2431697 association. It can also serve as a model for future studies of non-coding variants.

The following clarifications are needed:

- Sup Table 1: cohort tables"

The "total" do not correspond to the sum of the rows above. Further, the total genotyped + imputed do not correspond to the text (682 vs. 516). Please explain.

- L. 226: "robust association of rs2431697 ($P < 1.89E-22$)" but sup. Fig 2 shows meta $p = 1.04 \times 10^{-21}$

- Sup Fig. 6 not called in the text?

- Are some of the DEG in the rs2431697-KO cells (Fig. 2h) direct targets of miR-146a? The text mentions that the top pathway is TNF signaling, with enrichment for IL-17 and NF- κ B signaling. A more detailed analysis and reporting of the results would be useful.

- Fig. 3h-j: and Fig. 6 f-g: shouldn't the results analyzed as paired samples (WT and edited from the same donor)?

- Sup. Fig. 9: legend should indicate that GFP+ indicates cells transfected with the Adeno Cas9-GFP

- L. 421-423: miR-146a as a ncRNA does not have exons by definition.

Reviewer #3 (Remarks to the Author):

The manuscript "SLE non-coding genetic risk variant determines the epigenetic dysfunction of an immune cell specific enhancer that controls disease-critical microRNA expression" investigated potential regulatory function of SLE snp by using cutting-edge technology, CRISPR, based methods. Expression and fine regulation of microRNAs are important and dysregulation of microRNAs is implicated in many inflammatory diseases, but the regulatory mechanism is not clearly known. Major finding of the study is identification and confirmation risk variant rs2431369 as a causal variant for SLE by using several cutting-edge technologies. Author made several interesting observations: 1) rs2431369 allele specific expression of miR-146a, 2) risk variant is a potential distal promoter, 3) function of risk variant for miR-146a expression is monocyte specific (not T or B cells), and 4) risk variant-mediated miR-146a expression might be closely related to SLE pathogenesis by regulation of type I IFN expression.

Observations are interesting and most studies and data presented in the manuscript are convincing. Moreover, authors employed various up-to-dated technologies to prove conclusions, in vivo and in vitro studies. Some concerns should be addressed before publication.

1. Controversial data were observed in fig 1b and fig 3, primary B cells vs. Raji B cell line. Author needs to discuss why 30bp deletion clone in Raji have increased miR146 level and association of histone marks but not in primary B cells. It could be cell line specific phenomenon or transfection efficiency (primary B cells are extremely difficult to modify in vitro).

2. One of major limitation is figure 3 (and related sup. Figure 9). Calculation and confirmation of mutation by indel is common technique utilized in CRISPR technology. However, indel confirmation without sequencing does not guarantee whether monocyte, B or T cells have same mutation or not. Also, based on flow image in Sup. fig 9d, author should discuss the reason why monocytes are abnormally low in engrafted animals. Also, GFP gating in monocyte is different to the gating of T/B cells, If gating moves down similar to T/B, monocytes have more than 10% GFP-positivity. This

implies different transfection efficiency (and different modification rates in different cell types..).

3. Fig. 6e-g: Experimental protocol is not fully described. How IFN score is calculated? Author described the calculation method, but it is not clear why they use average (or combined) of multiple IFN signature genes. What about the expression of type I IFN? Also, monocytes are minor population among PBMCs and its percentage is variable depending on donor (5-15% of PBMCs). Therefore, the current data from total PBMC could be mitigated by dilution effects. And, the data presented is also not very impressive and much less significant compared to other data (from the isolated cells). And, what is "high IFN score" ? What is speculation of the effect on moderate-low IFN score patients?

4. What about miR-146a expression level in monocytes from SLE patients with T allele compared to C allele?

Reviewer #1 (Remarks to the Author):

Hou/Harley et al combine experimental and computational approaches to pinpoint the function of an SLE-associated non-coding variant to an enhancer controlling a microRNA expression. The microRNA in question, miRNA-146a, has a known role in inflammation and was already hypothesised (based on limited evidence) to underpin SLE associations in this locus. Nonetheless, the authors validate this association in an impressively comprehensive way, which probably makes their contribution worth seeing light in a high-impact journal such as Nat Comms.

Technical comments:

1. Lines 433-441:

Can the authors please show the actual quantitative data for 4C-seq beyond the CIRCOS plots.

Response: We have added the actual quantitative data (Figure R1) in the revised manuscript (Fig. 4h in the revised manuscript).

Figure R1. Contact profiles of the rs2431697 SNP site (top panel) and Pri-miR-146 promoter (bottom panel) using a 2-kb window size in main trend subpanel (black line). Red arrow head indicated view point position. Gray dots indicate normalized contact intensities and gray band shows the 20-80% percentiles. Heat map displays a set of medians of normalized contact intensities calculated at different window sizes (from 2 kb to 50 kb).

2. Lines 462-471:

My understanding is that at present, mass spectrometry-based identification of factors binding to a given non-repetitive DNA sequence remains highly challenging, with potentially noisy results. Is this why the data to this extent is confined to a supplementary figure? I would like the authors to elaborate more on how this method was performed and its limitations.

Response: Based on our MS result, it's actually a challenge to definite the binding proteins of targeting DNA sequence. We originally planned to cut the specific band after Coomassie Blue staining to perform MS. Unfortunately, we cannot observe the specific band and decide to cut the whole lane to perform MS study. In our study, a 41-bp DNA sequence can pull down more than 100 proteins, these proteins include histone proteins, ribosomal proteins, chromatin structure maintenance related proteins, transcription factors and so on. It's really difficult to distinguish the real binding and functional proteins. Given this, we set some criteria to filter the candidate proteins. The criteria are as follows: 1. proteins must be detected only in experiment group; 2. proteins must be detected in two replicate studies; 3. proteins should be transcription factors since our group found transcription factors (TFs) occupy multiple loci associated with complex genetic disorders¹. Based on above criteria, we identified the proteins in the supplementary figure. For this part, we have expanded the method in the revised manuscript.

3. Lines 497-504:

a) I am a little confused as to how the MARIO allele-specific analysis works. What exactly is the null model? Did I get it right that it's a non-parametric method?

Response: Yes, MARIO is indeed a non-parametric method. At its heart is the calculation of the "allelic reproducibility score". This metric takes multiple considerations into account, including the amount of allelic imbalance, the number of reads, and agreement across replicates. We decided to make our own method because, at the time, existing methods had prohibitively long running times, could not handle replicates, and/or did not let the user provide the location of heterozygotes as input (instead, they used the reads themselves to identify hits, which we have found is problematic in practice).

More details on the method are available in the original publication: Transcription factors operate across disease loci, with EBNA2 implicated in autoimmunity. Harley JB, Chen X, Pujato M, Miller D, Maddox A, Forney C, Magnusen AF, Lynch A, Chetal K, Yukawa M, Barski A, Salomonis N, Kaufman KM, Kottyan LC, Weirauch MT. Nat Genet. 2018 May;50(5):699-707. doi: 10.1038/s41588-018-0102-3. Epub 2018 Apr 16. PMID: 29662164

b) Did the authors benchmark their findings against WASP and/or RASQUAL?
This would be very welcome.

Response: Several years ago, we benchmarked MARIO against the ABC method², which also identifies allelic read imbalance. We compared the 19,871 scores produced by MARIO and ABC across 89 different GM12878 ChIP-seq datasets, and observed strong agreement between the two methods (Spearman correlation of 0.79, $P < 10^{-15}$).

We have also developed a procedure for gauging agreement between allelic ChIP-seq read imbalance and allelic TF binding predictions based on TF binding motifs. Due to numerous factors influencing TF binding, we do not expect 100% concordance between TF motif predictions and in vivo binding patterns – for example, reliance on binding co-factors, chromatin state, local DNA topology, and independence assumptions made by the PWM³ (see Slattery et al. 2014; PMID: 25129887 for a nice review). In brief, we first consider each heterozygous variant located within a peak in a given ChIP-seq experiment. We restrict to the subset of these with strong allelic read imbalance (MARIO ARS values > 0.5). We further restrict to those variants located within a predicted TF binding site for the TF for which the ChIP-seq experiment was performed (using a PWM cutoff of 0.50), and note the allele with the stronger predicted binding site score. We then calculate the counts for each of the four possible relationships between the reads and the motif scores (Figure R2). Higher counts for N1 and N4 indicate agreement between the two independent methods. The resulting table results in a χ^2 statistic for each ChIP-seq experiment. In total, ten TFs had at least 50 loci available for this analysis. Among these, five showed significant agreement ($P < 0.05$) between the allelic ChIP-seq reads and motif scores, including CTCF, which was the most significant ($P < 10^{-5}$), and is also the same TF for which this approach was applied in the RASQUAL paper⁴. Thus, this completely independent method validates MARIO's ARS values. Since the MARIO method is already published, and we are actively working on a “flagship paper” describing MARIO and applying it to the entire GEO database, we would prefer not including these validations in the present manuscript.

	More Reads (Ref)	More Reads (Non-Ref)
Better Motif (Ref)	N ₁	N ₂
Better Motif (Non-Ref)	N ₃	N ₄

Figure R2. Schematic for gauging the agreement between allelic ChIP-seq reads and predicted allelic TF binding. See text for discussion.

c) Could the authors elaborate why allele-specific signal was only formally detected in CHIP-seq data but not in ATAC-seq and FAIRE-seq?

Response: We actually did observe allele-dependent behavior in our own ATAC-seq data – see Figure S10d. Further, the data all agree, with ATAC-seq (Fig. S10d), three different histone marks from four different datasets (Fig. S10e) all strongly preferring the “C” allele, which is the allele with higher miR-146a expression (Fig. 5b), stronger NF-κB binding (Fig. 5c, d, e) and FAIRE signal (Fig. 5f).

4. Could the authors elaborate on the direct targets of miRNA-146a?

Response: Yes. Thank you for pointing out this oversight in our discussion. We have expanded the text to address this point. Several validated direct targets of miRNA-146a of potential relevance to SLE pathophysiology have been previously reported^{5,6}. These targets include: IRF5, STAT1, IRAK1 and TRAF6. Notably, all of these genes have been reported to be genes mediating biologic risk within SLE genetic association intervals⁷⁻¹⁰. This, of course adds both complexity and further supports to the role of miRNA-146a in SLE risk.

Are any of them already known and validated?

Response: Our group⁵ and other groups⁶ have validated that SLE risk genes IRF5, STAT1, IRAK1 and TRAF6 are the miR-146a target by Western Blot or luciferase reporter assay (Figure R3).

Figure R3. miR-146a targets identified by WB and luciferase reporter assay in the literature^{5,6}. (A, E-F) Sequence alignment of miR-146a and its target sites in 3'-UTR of IRF5, STAT1, TRAF6 and IRAK1. (B-D) Luciferase reporter assay and WB identified IRF5 and STAT1 are the direct targets of miR-146a. (G-H) Luciferase reporter assay identified TRAF6 and IRAK1 are the direct targets of miR-146a.

If not, could the authors at least run some target prediction algorithms and report the results?

Response: We have done this with TargetScan (microRNA target scan prediction) miRDB (miRNA target prediction) and MiRTarBase (a public database of experimentally validated microRNA targets), the number of miR-146a prediction targets is 283, 44 and 488, respectively. There are five

overlapping genes (ERBB4, IRAK1, TRAF6, CARD10, NUMB) among three groups (Figure R4) and we have provided this analysis in supplementary Table.

Figure R4. A Venn diagram showing overlapping genes of miR-146a prediction targets among TargetScan, miRtarBase and miRDB groups.

Minor textual comments:

Line 479: Suggest mentioning explicitly that RELA is an NFkB subunit.

Response: Thank you for your suggestion. We have described RELA as an NFkB subunit in the revised manuscript.

Lines 510-515: This sentence is a bit too long and convoluted. Suggest splitting it into two or more to take the reader through the rationale for this experiment a bit more slowly and clearly.

Response: Thank you for your suggestion. We have revised the sentence to more clearly describe the experiment in the revised manuscript.

References:

1. Harley, J.B. *et al.* Transcription factors operate across disease loci, with EBNA2 implicated in autoimmunity. *Nat Genet* **50**, 699-707 (2018).
2. Bailey, S.D., Virtanen, C., Haibe-Kains, B. & Lupien, M. ABC: a tool to identify SNVs causing

- allele-specific transcription factor binding from CHIP-Seq experiments. *Bioinformatics* **31**, 3057-9 (2015).
3. Slattery, M. *et al.* Absence of a simple code: how transcription factors read the genome. *Trends Biochem Sci* **39**, 381-99 (2014).
 4. Kumasaka, N., Knights, A.J. & Gaffney, D.J. Fine-mapping cellular QTLs with RASQUAL and ATAC-seq. *Nat Genet* **48**, 206-13 (2016).
 5. Tang, Y. *et al.* MicroRNA-146A contributes to abnormal activation of the type I interferon pathway in human lupus by targeting the key signaling proteins. *Arthritis Rheum* **60**, 1065-75 (2009).
 6. Taganov, K.D., Boldin, M.P., Chang, K.J. & Baltimore, D. NF-kappaB-dependent induction of microRNA miR-146, an inhibitor targeted to signaling proteins of innate immune responses. *Proc Natl Acad Sci U S A* **103**, 12481-6 (2006).
 7. Kottyan, L.C. *et al.* The IRF5-TNPO3 association with systemic lupus erythematosus has two components that other autoimmune disorders variably share. *Hum Mol Genet* **24**, 582-96 (2015).
 8. Patel, Z.H. *et al.* A plausibly causal functional lupus-associated risk variant in the STAT1-STAT4 locus. *Hum Mol Genet* **27**, 2392-2404 (2018).
 9. Kaufman, K.M. *et al.* Fine mapping of Xq28: both MECP2 and IRAK1 contribute to risk for systemic lupus erythematosus in multiple ancestral groups. *Ann Rheum Dis* **72**, 437-44 (2013).
 10. Namjou, B. *et al.* Evaluation of TRAF6 in a large multiancestral lupus cohort. *Arthritis Rheum* **64**, 1960-9 (2012).

Reviewer #2 (Remarks to the Author):

This study combines genomic, genetic, epigenetics and mechanistic assays to show that rs2431697 is a causal SNP that regulates miR-146a expression with the T/A allele being protective in a cell-specific manner. The authors show convincingly that this SNP corresponds to an open chromatin, epigenetically marked enhancer region of miR-146a in monocytes, and that the risk allele binds NF- κ B more than the protective allele. Moreover, the last experiment shows that the deletion of the SNP region in monocytes from SLE patients lowers their high IFN score, providing a nice proof of principle of its causal effect in its association with SLE susceptibility. This study provides a definitive mechanism linking low miR-146a expression in SLE susceptibility and rs2431697 association. It can also serve as a model for future studies of non-coding variants.

The following clarifications are needed:

- Sup Table 1: cohort tables”

The “total” do not correspond to the sum of the rows above.

Response: Thank you for pointing this out. This was an error that was carried over from an earlier version of the manuscript/analysis, prior to the removal of those individuals from the discovery cohort who could potentially overlap with the replication cohort (described in the Quality Control & Sample Overlap section). The table has been revised to contain the correct totals.

Further, the total genotyped + imputed do not correspond to the text (682 vs. 516). Please explain.

Response: Thank you for pointing this out. The way this information was presented was confusing. In the prior version, the “Imputed” column contained the number of additional markers in the imputation analysis that were not genotyped. However, the “Total” row included the total number of unique variants (whether genotyped or imputed) in the final imputation analysis (whether genotyped or imputed) – 608 unique variants altogether. We have revised the table to include an additional “total variants” column and hopefully this improves its clarity. We have also revised the text of the manuscript to contain the correct number of variants – 517, not 516. This was a typo.

- I. 226: “robust association of rs2431697 ($P < 1.89E-22$)” but sup. Fig 2 shows meta $p = 1.04 \times 10^{-21}$

Response: Thank you for pointing this out. The prior difference between the P-value called in the text and the P-value in the figure was due to the default behavior of the Forest Plot software (METASOFT/PMForestPlot – <http://genetics.cs.ucla.edu/meta/>) to display Meta-analysis results based on an inverse variance approach and not using the sample size weighted meta-analysis that we performed using METAL¹. This figure (and supplemental Figure 2 and 3) have been updated accordingly and the summary P-values now reflect the sample size weighted meta-analysis approach as implemented in METAL.

- Sup Fig. 6 not called in the text?

Response: Thank you for pointing this out. We have revised the text.

- Are some of the DEG in the rs2431697-KO cells (Fig. 2h) direct targets of miR-146a? The text mentions that the top pathway is TNF signaling, with enrichment for IL-17 and NF-kB signaling. A more detailed analysis and reporting of the results would be useful.

Response: Several validated direct targets of miRNA-146a of potential relevance to SLE pathophysiology have been previously reported^{2,3}. These targets include: IRF5, STAT1, IRAK1 and TRAF6. However, these genes are not in the DEG list. Actually, this result is not surprising. It is known that microRNAs regulate gene expression by promoting mRNA degradation or inhibiting mRNA translation, thereby reducing the levels of protein. Interestingly, these targets are initially validated by WB or luciferase reporter assay in protein level rather than RNA level by over-expressing miR-146a, so we think protein level change is the direct evidence that they are miR-146a targets. Meanwhile, a typical miRNA-target interaction only produces a slight reduction of most targeted proteins^{4,5}. For example, miR-155 overexpression only has mild effects (~20% to 30%) on the synthesis of most of the 3,000-3500 proteins in HeLa cells⁶ assayed by a mass spectrometry-based pSILAC (pulsed stable isotope labeling with amino acids in cell culture) method. However, the slight reduction of a certain protein can be reflected by its downstream gene expression. To analyze the downstream gene expression of STAT1 (a key regulator of the type I Interferon pathway⁷), IRF5, IRAK1 and TRAF6 (key regulators in proinflammatory cytokine production⁸), we performed RNA sequencing using rs2431697 WT clones and KO clones after IFN- α stimulation or TNF- α stimulation. Then, we use QIAGEN'S Ingenuity Pathway Analysis (IPA), a web-based software application that is broadly adopted in the life science community and has been cited in thousands of peer-reviewed journal articles, to predict the downstream genes or network-regulated genes of STAT1, IRF5, IRAK1 and TRAF6 from the RNA-seq data. After that, we compared the expression of these genes in WT group and KO group. As shown in Figure R5A-

D, most of the downstream genes or network-regulated genes are up-regulated in the KO group. In addition, some of these genes were further validated by RT-qPCR (Figure R5E-H). Thus, we believe these miR-146a target genes are actually changed in the KO group.

For the pathway analysis, we have included an additional supplementary table, with links to Enrichr (<https://amp.pharm.mssm.edu/Enrichr/>) pathway analyses of the genes DEG between rs2431697KO and rs2431697WT cells as well as target scan predictions, mirTarBase and miRDB targets for human mir-146a-5p.

Figure R5. (A-D) Downstream genes or network-regulated genes of TRAF6, IRAK1, STAT1 and IRF5 analyzed by IPA are up-regulated in KO group

compared with WT group indicating by heat map. (E-H) RT-qPCR validated the differentially expressed genes of miR-146a direct targets in different clones. Data are represented as mean \pm SEM, and p values are calculated using unpaired Student's t test. *p < 0.05; **p < 0.01; ***p < 0.001.

- Fig. 3h-j: and Fig. 6 f-g: shouldn't the results analyzed as paired samples (WT and edited from the same donor)?

Response: Thank you for pointing out this issue, and we have re-analyzed the data and revised in the manuscript.

- Sup. Fig. 9: legend should indicate that GFP+ indicates cells transfected with the Adeno Cas9-GFP

Response: Thank you for your suggestion. We have revised the legend in the manuscript.

- L. 421-423: miR-146a as a ncRNA does not have exons by definition.

Response: Certainly, the processed miR-146a does not. However, that the host gene (MIR3142HG), encoding pri-miR-146a undergoes RNA splicing, so has introns (Figure R6) and the two portions that are spliced together have previously been referred to as exons³. We have revised the text in an attempt at greater precision in this area.

Figure R6. The information of miR-146a's host gene-MIR3142HG from UCSC.

References:

1. Willer, C.J., Li, Y. & Abecasis, G.R. METAL: fast and efficient meta-analysis of genomewide association scans. *Bioinformatics* **26**, 2190-1 (2010).
2. Tang, Y. *et al.* MicroRNA-146A contributes to abnormal activation of the type I interferon pathway in human lupus by targeting the key signaling proteins. *Arthritis Rheum* **60**, 1065-75 (2009).
3. Taganov, K.D., Boldin, M.P., Chang, K.J. & Baltimore, D. NF-kappaB-dependent induction of microRNA miR-146, an inhibitor targeted to signaling proteins of innate immune responses. *Proc Natl Acad Sci U S A* **103**, 12481-6 (2006).
4. Baek, D. *et al.* The impact of microRNAs on protein output. *Nature* **455**, 64-71 (2008).
5. Schmiedel, J.M. *et al.* Gene expression. MicroRNA control of protein expression noise.

- Science* **348**, 128-32 (2015).
6. Selbach, M. *et al.* Widespread changes in protein synthesis induced by microRNAs. *Nature* **455**, 58-63 (2008).
 7. Ivashkiv, L.B. & Donlin, L.T. Regulation of type I interferon responses. *Nat Rev Immunol* **14**, 36-49 (2014).
 8. Lazzari, E. & Jefferies, C.A. IRF5-mediated signaling and implications for SLE. *Clin Immunol* **153**, 343-52 (2014).

Reviewer #3 (Remarks to the Author):

The manuscript “SLE non-coding genetic risk variant determines the epigenetic dysfunction of an immune cell specific enhancer that controls disease-critical microRNA expression” investigated potential regulatory function of SLE snp by using cutting-edge technology, CRISPR, based methods. Expression and fine regulation of microRNAs are important and dysregulation of microRNAs is implicated in many inflammatory diseases, but the regulatory mechanism is not clearly known. Major finding of the study is identification and confirmation risk variant rs24313697 as a causal variant for SLE by using several cutting-edge technologies. Author made several interesting observations: 1) rs24313697 allele specific expression of miR-146a, 2) risk variant is a potential distal promoter, 3) function of risk variant for miR-146a expression is monocyte specific (not T o B cells), and 4) risk variant-mediated miR-146a expression might be closely related to SLE pathogenesis by regulation of type I IFN expression.

Observations are interesting and most studies and data presented in the manuscript are convincing. Moreover, authors employed various up-to-dated technologies to prove conclusions, in vivo and in vitro studies. Some concerns should be addressed before publication.

1. Controversial data were observed in fig 1b and fig 3, primary B cells vs. Raji B cell line. Author needs to discuss why 30bp deletion clone in Raji have increased miR146 level and association of histone marks but not in primary B cells. It could be cell line specific phenomenon or transfection efficiency (primary B cells are extremely difficult to modify in vitro).

Response: We thank the reviewer for pointing out our lack of clarity on this issue. We suspect that the rs2431697 miR-146a enhancer is likely operative in B-cells, but only under certain circumstances. In primary B cells, rs2431697-containing region is only enriched with H3K4me1 modification but without H3K27ac signal (Fig. 1b). Meanwhile, ATAC-seq analysis indicates this region is closed in primary B cells (Fig. 1b). This indicates the region of rs2431697 is a poised enhancer in primary B cells, which is directly validated by the result that disruption of this region by CRISPR has no effect on miR-146a expression (Fig.3i and 3m). In RAJI cells, rs2431697-containing region is an active enhancer reflecting by high signal of H3K4me1 and H3K27ac modification (Fig. 3e) and high chromatin accessibility (Fig. 3f), we can observe that deletion of 30-bp fragment harboring rs2431697 decreased miR-146a expression in Raji cells (Fig. 3a). Indeed, in addition to being an immortalized cell line, Raji cells harbor Epstein-Barr Virus (<https://www.atcc.org/products/all/CCL-86.aspx>), so we suspect EBV infection changes the status of this region and may have important roles in B-cell

biology. To test this hypothesis, we analyzed the chromatin landscape of rs2431697-containing region in GM12878 cells (EBV transformed B cell line), and observed high H3K27ac and H3K4me1 signal enriched in rs2431697-containing region (Figure R7A). Further, we also demonstrate this region marks with strong H3K4me1 and H3K27ac signal (two active enhancer markers) in another EBV transformed B cell line (LCL1) constructed by our lab (Figure R7B). More importantly, we disrupted the rs2431697-containing region in the LCL1 cells, resulting in a mixture of various genotypes, not a single cell clone with definite genotype, and observed the regulation of rs2431697-containing region on miR-146a expression (Figure R7C). Based on the above discussion, we suspect that context of EBV infection can activate this enhancer in primary B cells and active investigation into these and other SLE-relevant B-cell stimuli is ongoing and will be published separately. We have expanded our discussion of this point in the text.

As pointed out, primary B-cells are notoriously difficult to modify in vitro. However, for the cells under study, the T7EI assay indicates similar efficiency of deletion in all of the primary cells using Cas9 RNP electroporation (Fig. S9a-c), a technique that has recently been developed to effectively, though not especially efficiently, alter the genome of primary human B-cells^{1,2}. Because of this, we do not suspect that efficiency is the reason we observe changes in miR-146a expression. A cell-line specific phenomenon is certainly possible.

A

B

Figure R7. Enhancer of rs2431697-containing region regulating miR-146a expression may be associated with EBV infection in B cells. (A) H3K4me1 and H3K27ac signal of rs2431697-containing region in GM12878 cell line (EBV transformed B cells) from UCSC. (B) CHIP-qPCR analyzed enhancer signals in LCL1 (An EBV transformed B cells in our lab). (C) CRISPR/Cas9 disrupted rs2431697-containing region decreases miR-146a expression in LCL1 cells. Data are represented as mean \pm SEM, and p values are calculated using unpaired Student's t test. * $p < 0.05$; ** $p < 0.01$; *** $p < 0.001$.

2. One of major limitation is figure 3 (and related sup. Figure 9). Calculation and confirmation of mutation by indel is common technique utilized in CRISPR technology. However, indel confirmation without sequencing does not guarantee whether monocyte, B or T cells have same mutation or not.

Response: To identify the sequences present in the edited cells, we first amplified the targeting locus harboring the editing region by specific primers. Then PCR products were sequenced by Sanger sequence, and the data of

Sanger sequencing was analyzed with ICE (Inference of CRISPR Edit) (<https://ice.synthego.com/#/>)³, which can quantify each editing outcome observed in the mixed Sanger read and the analyzed results correlate well with next-generation sequencing of amplicons (Amp-Seq) (Figure R8).

Figure R8. ICE analysis flow scheme (A) and correlation with next-generation sequencing of amplicons (B-C).

As shown in Figure R9-R11, all samples were efficiently edited, especially in the primary T cells. Although the edited cells are a heterogeneous population with various mutations, the proportion of 30-bp deletion accounts for majority in the edited population.

RELATIVE CONTRIBUTION OF EACH SEQUENCE (NORMALIZED)

RELATIVE CONTRIBUTION OF EACH SEQUENCE (NORMALIZED)

RELATIVE CONTRIBUTION OF EACH SEQUENCE (NORMALIZED)

RELATIVE CONTRIBUTION OF EACH SEQUENCE (NORMALIZED)

RELATIVE CONTRIBUTION OF EACH SEQUENCE (NORMALIZED)

Figure R9. Indel frequency in the edited primary T cells in vitro by ICE analysis.

Figure R10. Indel frequency in the edited primary B cells in vitro by ICE analysis.

INDEL	CONTRIBUTION	SEQUENCE
0	42%	GAAAGAGGAAGCTCAGTGCCACATGA
-30	32%	GAAAGAGGAAGCTCAGTGCCACATGA
-6	3%	GAAAGAGGAAGCTCAGTGCCACATGA
-5	2%	GAAAGAGGAAGCTCAGTGCCACATGA
-1	2%	GTGGGGCTGAAATAAAAAACCTCGA
-6	1%	GAAAGAGGAAGCTCAGTGCCAC
-5	1%	GAAAGAGGAAGCTCAGTGCCAC
-1	1%	GAAAGAGGAAGCTCAGTGCCACATGA

INDEL	CONTRIBUTION	SEQUENCE
0	30%	GAAAGAGGAAGCTCAGTGCCACATGA
-30	2%	GAAAGAGGAAGCTCAGTGCCACATGA
-19	4%	GTGGGGCTGAAATAAAAAACCTCGA
-2	4%	GTGGGGCTGAAATAAAAAACCTCGA
-2	4%	GTGGGGCTGAAATAAAAAACCTCGA
-25	3%	GTGGGGCTGAAATAAAAAACCTCGA
-30	2%	GAAAGAGGAAGCTCAGTGCCACATGA
-20	2%	GTGGGGCTGAAATAAAAAACCTCGA
-8	2%	GTGGGGCTGAAATAAAAAACCTCGA
-14	1%	GTGGGGCTGAAATAAAAAACCTCGA
-14	1%	GTGGGGCTGAAATAAAAAACCTCGA
-10	1%	GTGGGGCTGAAATAAAAAACCTCGA

INDEL	CONTRIBUTION	SEQUENCE
0	48%	GAAAGAGGAAGCTCAGTGCCACATGA
-30	20%	GAAAGAGGAAGCTCAGTGCCACATGA
-5	4%	GAAAGAGGAAGCTCAGTGCCACATGA
-6	3%	GAAAGAGGAAGCTCAGTGCCACATGA
-3	3%	GAAAGAGGAAGCTCAGTGCCACATGA
-7	2%	GAAAGAGGAAGCTCAGTGCCACATGA
-2	2%	GAAAGAGGAAGCTCAGTGCCACATGA
-36	1%	GAAAGAGGAAGCTCAGTGCCAC
-6	1%	GAAAGAGGAAGCTCAGTGCCACATGA

INDEL	CONTRIBUTION	SEQUENCE
0	32%	GAAAGAGGAAGCTCAGTGCCACATGA
-30	22%	GAAAGAGGAAGCTCAGTGCCACATGA
-6	8%	GTGGGGCTGAAATAAAAAACCTCGA
-5	6%	GAAAGAGGAAGCTCAGTGCCACATGA
-1	3%	GAAAGAGGAAGCTCAGTGCCACATGA
-11	2%	GAAAGAGGAAGCTCAGTGCCACATGA
-32	1%	GAAAGAGGAAGCTCAGTGCCAC
-12	1%	GAAAGAGGAAGCTCAGTGCCACATGA
-12	1%	GAAAGAGGAAGCTCAGTGCCACATGA
-1	1%	GAAAGAGGAAGCTCAGTGCCACATGA
-19	1%	GTGGGGCTGAAATAAAAAACCTCGA
-19	1%	GTGGGGCTGAAATAAAAAACCTCGA
-6	1%	GAAAGAGGAAGCTCAGTGCCAC

INDEL	CONTRIBUTION	SEQUENCE
0	43%	GAAAGAGGAAGCTCAGTGCCACATGA
-30	17%	GAAAGAGGAAGCTCAGTGCCACATGA
-7	4%	GAAAGAGGAAGCTCAGTGCCACATGA
-3	3%	GAAAGAGGAAGCTCAGTGCCACATGA
-6	3%	GAAAGAGGAAGCTCAGTGCCACATGA
-5	2%	GAAAGAGGAAGCTCAGTGCCACATGA
-3	2%	GAAAGAGGAAGCTCAGTGCCACATGA
-13	1%	GAAAGAGGAAGCTCAGTGCCACATGA
-13	1%	GAAAGAGGAAGCTCAGTGCCACATGA
-5	1%	GAAAGAGGAAGCTCAGTGCCACATGA
-4	1%	GAAAGAGGAAGCTCAGTGCCACATGA
-4	1%	GAAAGAGGAAGCTCAGTGCCACATGA
-5	1%	GAAAGAGGAAGCTCAGTGCCACATGA

Figure R11. Indel frequency in the edited primary monocytes in vitro by ICE analysis.

Also, based on flow image in Sup. fig 9d, author should discuss the reason why monocytes are abnormally low in engrafted animals.

Response: There are some references indicating that in human PBMCs reconstituted humanized mice, T cell engraftment predominates and the proportion of monocytes is relatively low. Meanwhile, the degree of engraftment is time dependent, and different from model-to-model, donor-to-donor. Palamides *etal.*⁴ developed a disease mouse model that relies on NSG mice reconstituted with PBMCs of UC-affected individuals. In their model, the percentage of CD14+ cells in human leucocytes isolated from mouse spleen is about 2.78% (Figure R12A). Consistent with this result, other studies⁵⁻⁸ that use human PBMCs reconstitute humanized mice also found the proportion of CD14+ cells in human CD45+ cells is less than 5% (Figure R12B-E). Based on the discussion above, we think the defect of this model results in the low level of monocytes in engrafted animals.

Figure R12. Percentage of CD14+ cells in human leucocytes in human PBMCs reconstituted humanized mouse from literature. A from Palamides, P. *et al.*⁴, B from Palamides, Ye, C. *et al.*⁵, C from Fukasaku, Y. *et al.*⁷, D from Zhao, Y. *et al.*⁸, E from Zhou, J. *et al.*⁶.

Also, GFP gating in monocyte is different to the gating of T/B cells, If gating moves down similar to T/B, monocytes have more than 10% GFP-positivity. This implies different transfection efficiency (and different modification rates in different cell types.)

Response: We re-analyzed the FACS data of 3 samples using the same gating strategy to compare the GFP+ positive cells in different immune cell subsets. The percentage of CD14+ GFP+ cells in one sample reaches to 18.9%, whereas the other two samples are less than 1%. Besides, the GFP+ ratios in CD19+ B cells are around 1%. Although CD3+ T cells is the predominate cell type, the percentage of GFP+ are the lowest when compared with CD14+ monocytes and CD19+ B cells (Figure R13). Furthermore, we sorted the strong GFP+ cells of different immune cell subsets of 3 individuals in each group to test the editing efficiency by amplification of the rs2431697 locus using locus-specific primer. The PCR products were sequenced by Sanger sequencing, and the editing outcome was calculated with ICE (<https://ice.synthego.com/#/>)³. Consistent with the FACS assay of GFP+ cells in each subset, the results of the analysis (Figure R14-R16) indicate that the locus in cells can be edited with high efficiency using the current gating strategy. We speculate that the discrepancy leading to higher GFP fluorescence intensities in CD14+ cells and relatively lower intensities in CD3+/CD19+ cell populations is an artefact of the high levels of autofluorescence exhibited by many myeloid cell populations⁹.

Figure R13. FACS plots represent analysis the percentages of GFP+ cells in human CD19+ B cells, CD3+ T cells, and CD14+ monocytes of human PBMC reconstituted NSG mice (n=3).

Figure R14. Indel frequency in the edited primary CD3+GFP+ T cells in vivo by ICE analysis.

Figure R15. Indel frequency in the edited primary CD19+GFP+ B cells in vivo by ICE analysis.

Figure R16. Indel frequency in the edited primary CD14+GFP+ monocytes in vivo by ICE analysis.

3. Fig. 6e-g: Experimental protocol is not fully described.

Response: Thank you for pointing this out. We have expanded the method in the revised manuscript.

How IFN score is calculated?

Response: First, RNA was isolated and converted to cDNA, expression of several IFN genes was measured by RT-qPCR. Results were expressed in

relative expression calculated according to the formula: Relative expression = $2^{-(Ct \text{ test gene} - Ct \text{ GAPDH})}$. Next, the mean expression level of IFIT3 in the negative control group (Sample was transfected with negative control sgRNAs) was calculated and subtracted from the expression level of IFIT3 of each sample in negative control group and positive control group (Samples were transfected with positive control sgRNAs), and the remainder was divided by the SD value of IFIT3 in negative control group to standardize the gene expression level. The standardized expression levels of IFI27, OAS1 and LY6E were calculated in the same method. Finally, the 4 values were summed to get the IFN score for each sample.

The relevant portion of the methods has been expanded in the revised manuscript.

Author described the calculation method, but it is not clear why they use average (or combined) of multiple IFN signature genes. What about the expression of type I IFN?

Response: IFNs potentially regulate the transcription of up to 2000 genes in an IFN subtype, dose, cell type and stimulus dependent manner (<http://www.interferome.org/interferome/home.jsp>). So, it's difficult to use single gene expression to evaluate the activation of type I IFN pathway. To better quantify type I IFN activity, we choose the common IFN-inducible genes based on the profiling data from SLE patients' PBMC (Figure R17)¹⁰⁻¹³ and use a composite score/signature integrating expression of these genes when dealing with SLE PBMC. More importantly, the IFN gene expression signature calculated based on this method has been served as a marker for more severe disease involving the kidneys, hematopoietic cells, and or the central nervous system (Figure R19)¹⁰.

For the expression of type I IFN, many profiling studies in SLE patients' PBMCs using microarray¹⁰⁻¹³ or RNA sequencing¹⁴ have found that type I IFNs expression cannot be detected or expressed at a very low level, especially for the IFN alpha. Further, the direct measurement of type I IFN protein in biological samples has remained elusive and current ELISAs have proven either insensitive or unreliable¹⁵. For example, Baechler¹⁰ *et al.* detected IFN-alpha protein expression in only 2 of 38 patients and 1 of 14 controls using ELISA method. This may be resulted by the short half-life of type I IFN or other reasons. Thus, the IFN gene expression signature in blood cells of patients appears to be a more sensitive readout for activation of this pathway than cytokine levels in serum.

Figure R17. IFN signature in SLE patients. (A) Gene expression profiles of PBMCs from 48 SLE patients and 42 healthy controls. Shown are hierarchical clustering results of microarray data for 161 genes. Red indicates genes expressed at higher levels relative to the control mean, and green represents genes expressed at lower levels than control mean. Black bars on the left side of the figure indicate IFN-regulated genes¹⁰. (B) Active SLE patients leukocytes (left panel) display 36 IFN-up-regulated and 13 down-regulated transcript sequences. The same genes are altered in healthy PBMCs cultured in vitro with IFN- α (right panel)¹¹. (C) Clustering of 25 SLE-associated genes from the microarray analysis. A cluster of IFN-induced genes is highlighted¹².

Also, monocytes are minor population among PBMCs and its percentage is variable depending on donor (5-15% of PBMCs). Therefore, the current data from total PBMC could be mitigated by dilution effects. And, the data presented is also not very impressive and much less significant compared to other data (from the isolated cells).

Response: Indeed, this is true. We isolated monocytes from SLE patients and performed CRISPRa assay targeting rs2431697 region, in this experiment, the average expression level of pri-miR-146a increased 2.77-fold in PC group compared the expression level in NC group, meanwhile the average of IFN score in PC group was about 332-fold lower than the data in NC group. This data was 1.3-fold and 4.7-fold change in total PBMCs experiment, respectively. This indicates a strong regulatory effect of rs2431697-containing region on miR-146a and interferon pathway in isolated monocytes compared the data in total PBMCs (Figure R18A-B).

The application of CRISPR/Cas9 technology for functional genomic study can fall into two categories: One that changes the targeted DNA sequence by CRISPR-Cas9 mediated fragment deletion, HDR or base editing, another that do not change the targeted DNA sequence by fusion activation proteins or repression proteins to dCas9 and interference gene expression by sgRNAs targeting the regulatory region. Deletion of an enhancer region is the gold standard to reveal the role of enhancer in gene expression regulation.

For the studies in the isolated primary cells from PBMCs, cells were transfected with Cas9 RNP using NEON system to cut the 30 bp fragment containing rs2431697 to study whether this enhancer region can regulate miR-146a expression in cell-type dependent manner (Figure R18C). Primary cells can be edited with high editing efficiency with Cas9 RNP (Figure R11), which results in the significant difference between control group and experiment group in monocytes.

For the CRISPR activation (CRISPRa) studies, this system could interfere with gene expression by targeting a regulatory region using specific sgRNA, but without altering the DNA sequence (Figure R18D). Although it has been widely used to study the function of regulatory elements, this system also has some disadvantages. The CRISPRa system is composed of three components: dCas9-vp64, sgRNA with modified scaffold and MS2-p65-HSF1 fusion protein (Figure R18E). The plasmids expressing these components are larger than 10 kb, and there are no commercial proteins. Coupled with the larger plasmids this results in lower cell viability and transfection efficiency^{16,17}, especially in primary cells. Moreover, T cells (and to a lesser extent B cells) account for the majority of human PBMCs. We suspect that substantial amounts of CRISPRa plasmids were transfected into the T cells and B cells rather than monocytes (Figure R18F), and based on our result, rs2431697-containing region is a monocyte-specific enhancer that can function in monocyte rather than T cells and B cells, so we think the low transfection efficacy and proportion of monocytes mitigates our ability to observe this regulatory effect in bulk PBMCs as mentioned by reviewer.

Figure R18. CRISPRa therapy decreases the IFN score of SLE patients' Monocytes based on sgRNAs targeting the rs2431697 site. (A) CRISPR activation system up-regulated pri-miR-146a expression in isolated CD14+ monocytes from SLE patients. (B) CRISPR activation system down-regulated IFN-score in isolated CD14+ monocytes from SLE patients. (n=4, replicates represent biological samples from unique individuals). (C) Flow scheme of CRISPR-mediated fragment deletion in isolated CD14+ monocytes. (D) Working principle for CRISPR activation system based on rs2431697-containing region. (E) Three components of CRISPR activation system. (F) Flow scheme of CRISPR activation system in total PBMCs. Data are

represented as mean \pm SEM, and p values are calculated using paired Student's t test. *p < 0.05; **p < 0.01; ***p < 0.001.

And, what is "high IFN score"?

Response: Gene expression profiling analysis indicates that genes in IFN signaling pathway are over-expressed (IFN signature) in SLE patients, and IFN score was designed to measure the intensity of IFN signature by calculating the expression levels of genes in the IFN pathway for each patient and control^{10,11,18,19}. Many studies found IFN score was increased in approximately half of SLE patients and associated with severe of SLE. For instance, patients with high IFN score had a strongly higher number of SLE criteria than patients with low IFN score, and most patients in high-IFN score group had hematologic, renal or CNS involvement in their disease compared with those in low-IFN score group¹⁰ (Figure R19).

We have expanded our discussion of high IFN score in the methods section.

Figure R19. The IFN expression signature identifies a clinical subset of SLE patients with severe disease¹⁰. (A) IFN score is higher in SLE patients than in healthy control. (B) IFN score is significantly associated with the number of SLE disease criteria. (C) Patients were divided into two groups: IFN-high, the 24 patients with the highest IFN scores; and IFN-low, the 24 patients with the

lowest scores. The data compare the two groups for number of ACR criteria for SLE (minimum of 4 to establish the disease, maximum of 11). (D) The data compare the percent of patients in the IFN-high and IFN-low groups with ACR-defined criteria for renal and or CNS disease or hematologic involvement.

What is speculation of the effect on moderate-low IFN score patients?

Response: It is difficult to say. There is some evidence suggesting that the bulk of SLE patients exhibit high type I IFN score/signature at some point during the course of their disease, particularly during disease flares²⁰. So, while introducing CRISPR-based alteration of the PBMC from a low/moderate type I IFN score SLE patient would not be expected to greatly lower the already low interferon score, a similar approach (targeting the rs2431697-enhancer) may yet prove broadly applicable to SLE therapy.

4. What about miR-146a expression level in monocytes from SLE patients with T allele compared to C allele?

Response: We isolated 80 monocytes samples from SLE patients to test the miR-146a expression difference between the different genotypes. In these samples, the CC genotype accounts for 5%, the CT genotype accounts for 37.5%, and TT genotypes accounts for 57.5%. Unfortunately, we do not observe a miR-146a expression difference between the rs2431697 different genotypes (Figure R20). SLE patients often require chronic immunosuppressive therapy in order to prevent organ threatening-flares and death. Importantly, the most commonly used and most effective immunosuppressive therapy for SLE patients are glucocorticoids. Glucocorticoids are so profoundly effective in controlling SLE disease activity that the ability of a novel therapeutic modality to reduce the glucocorticoid burden is a widely applied secondary endpoint that argues for efficacy of novel SLE therapies^{21,22}. Importantly, a major mechanism of action of glucocorticoid therapy lies in their ability to interfere with NF-kB activity²³. We have previously reported that miR-146a expression changes with the various stages of disease²⁴. So, we suspect that both the treatment and stage of disease may influence miR-146a expression more powerfully than do the rs2431697 alleles, which results in our observing this negative result. Further, there were only 4 carriers for the homozygous CC genotype in total of 80 SLE patients, so even with 80 SLE patients, we are likely lack statistical power to detect a true difference. According to the percentage of CC genotype in Chinese SLE patients in our data, we estimate from a power analysis that we would need to increase the sample size to 20 rs2431697 CC genotype carriers to make the observed difference significant. To do this, we estimate that we would have to recruit an additional 320 patients, which is too

challenging for us at this time. In addition, we are planning to study the association between genotypes and gene expression in primary cells using prime editing technology, which could edit the allele and substitute with the any desirable base without excess byproducts²⁵. These planned experiments are anticipated to provide a very strong test of the hypothesis that rs2431697 regulates miR-146a expression in a cell-type dependent manner. Unfortunately, isolating these effects *in vitro* will be a major challenge and we anticipate that it will require a few years work.

Figure R20. miR-146a expression in SLE patients' monocytes with rs2431697 different alleles.

Last but not least, we thank the reviewer's positive evaluation of our manuscript and for the insightful comments and constructive suggestions to help us improve the manuscript. We are particularly appreciative as the review process happens during an unusual time. Thank you very much and best wishes to all.

References:

1. Wu, C.-A.M. *et al.* Genetic engineering in primary human B cells with CRISPR-Cas9 ribonucleoproteins. *Journal of Immunological Methods* **457**, 33-40 (2018).
2. Johnson, M.J., Laoharawee, K., Lahr, W.S., Webber, B.R. & Moriarity, B.S. Engineering of Primary Human B cells with CRISPR/Cas9 Targeted Nuclease. *Scientific Reports* **8**, 12144 (2018).
3. Hsiao, T. *et al.* Inference of CRISPR Edits from Sanger Trace Data. *bioRxiv*, 251082 (2018).
4. Palamides, P. *et al.* A mouse model for ulcerative colitis based on NOD-scid IL2R γ null mice reconstituted with peripheral blood mononuclear cells from affected individuals.
5. Ye, C. *et al.* A rapid, sensitive, and reproducible *in vivo* PBMC humanized murine model for determining therapeutic-related cytokine release syndrome. *The FASEB Journal* **34**, 12963-12975 (2020).
6. Zhou, J. *et al.* Anti-inflammatory Activity of MTL-CEBPA, a Small Activating RNA Drug, in LPS-Stimulated Monocytes and Humanized Mice. *Molecular Therapy* **27**, 999-1016 (2019).

7. Fukasaku, Y. *et al.* Novel immunological approach to assess donor reactivity of transplant recipients using a humanized mouse model. *Hum Immunol* **81**, 342-353 (2020).
8. Zhao, Y. *et al.* Development of a new patient-derived xenograft humanised mouse model to study human-specific tumour microenvironment and immunotherapy. *Gut* **67**, 1845-1854 (2018).
9. Mitchell, A.J. *et al.* Technical advance: autofluorescence as a tool for myeloid cell analysis. *J Leukoc Biol* **88**, 597-603 (2010).
10. Baechler, E.C. *et al.* Interferon-inducible gene expression signature in peripheral blood cells of patients with severe lupus. *Proc Natl Acad Sci U S A* **100**, 2610-5 (2003).
11. Bennett, L. *et al.* Interferon and granulopoiesis signatures in systemic lupus erythematosus blood. *J Exp Med* **197**, 711-23 (2003).
12. Ye, S. *et al.* Protein interaction for an interferon-inducible systemic lupus associated gene, IFIT1. *Rheumatology (Oxford)* **42**, 1155-63 (2003).
13. Han, G.M. *et al.* Analysis of gene expression profiles in human systemic lupus erythematosus using oligonucleotide microarray. *Genes Immun* **4**, 177-86 (2003).
14. Panwar, B. *et al.* Integrative transcriptomic analysis of SLE reveals IFN-driven cross-talk between immune cells. *bioRxiv*, 2020.04.27.065227 (2020).
15. Rodero, M.P. *et al.* Detection of interferon alpha protein reveals differential levels and cellular sources in disease. *The Journal of experimental medicine* **214**, 1547-1555 (2017).
16. Lesueur, L.L., Mir, L.M. & André, F.M. Overcoming the Specific Toxicity of Large Plasmids Electrotransfer in Primary Cells In Vitro. *Molecular therapy. Nucleic acids* **5**, e291-e291 (2016).
17. Søndergaard, J.N. *et al.* Successful delivery of large-size CRISPR/Cas9 vectors in hard-to-transfect human cells using small plasmids. *Communications Biology* **3**, 319 (2020).
18. Kirou, K.A. *et al.* Coordinate overexpression of interferon-alpha-induced genes in systemic lupus erythematosus. *Arthritis Rheum* **50**, 3958-67 (2004).
19. Obermoser, G. & Pascual, V. The interferon- α signature of systemic lupus erythematosus. *Lupus* **19**, 1012-1019 (2010).
20. Banchereau, R. *et al.* Personalized Immunomonitoring Uncovers Molecular Networks that Stratify Lupus Patients. *Cell* **165**, 551-65 (2016).
21. Binda, V. *et al.* Belimumab may decrease flare rate and allow glucocorticoid withdrawal in lupus nephritis (including dialysis and transplanted patient). *J Nephrol* (2020).
22. Furie, R. *et al.* A phase III, randomized, placebo-controlled study of belimumab, a monoclonal antibody that inhibits B lymphocyte stimulator, in patients with systemic lupus erythematosus. *Arthritis Rheum* **63**, 3918-30 (2011).
23. Franco, L.M. *et al.* Immune regulation by glucocorticoids can be linked to cell type-dependent transcriptional responses. *Journal of Experimental Medicine* **216**, 384-406 (2019).
24. Tang, Y. *et al.* MicroRNA-146A contributes to abnormal activation of the type I interferon pathway in human lupus by targeting the key signaling proteins. *Arthritis Rheum* **60**, 1065-75 (2009).
25. Anzalone, A.V. *et al.* Search-and-replace genome editing without double-strand breaks or donor DNA. *Nature* **576**, 149-157 (2019).

REVIEWERS' COMMENTS

Reviewer #1 (Remarks to the Author):

I thank the authors for the responses to my comments and amendments. I am happy with the revised version overall. My only request would be for the authors to explain the limitations of their mass-spec approach in the main text along the lines of the explanation given in the rebuttal letter (in response to my point 2).

Reviewer #2 (Remarks to the Author):

All concerns have been addressed

Reviewer #3 (Remarks to the Author):

Authors responded well to all my previous concerns and questions.

Reviewer #1 (Remarks to the Author):

I thank the authors for the responses to my comments and amendments. I am happy with the revised version overall. My only request would be for the authors to explain the limitations of their mass-spec approach in the main text along the lines of the explanation given in the rebuttal letter (in response to my point 2).

Response: Thanks for your suggestion. We have added the description in the revised manuscript. We deeply appreciate your constructive comments that greatly help improve the presentation of this manuscript.

Reviewer #2 (Remarks to the Author):

All concerns have been addressed

Response: Thanks for your positive evaluation of our manuscript and for the insightful comments and constructive suggestions to help us improve the manuscript.

Reviewer #3 (Remarks to the Author):

Authors responded well to all my previous concerns and questions.

Response: Thanks for your positive evaluation of our manuscript and for the insightful comments and constructive suggestions to help us improve the manuscript.